# Electrochemical and Optical Carbon Dots and Glassy Carbon Biosensors: A Review on Their Development and Applications in Early Cancer Detection

**DOI:** 10.3390/mi16020139

**Published:** 2025-01-25

**Authors:** Juana G. López, Mariana Muñoz, Valentina Arias, Valentina García, Paulo C. Calvo, Alejandro O. Ondo-Méndez, Diana C. Rodríguez-Burbano, Faruk Fonthal

**Affiliations:** 1Biomedical Engineering Research Group—GBIO, Universidad Autónoma de Occidente, Cali 760030, Colombia; juana.lopez@uao.edu.co (J.G.L.); mariana.munoz_valdes@uao.edu.co (M.M.); valentina.arias@uao.edu.co (V.A.); valentina.garcia_per@uao.edu.co (V.G.); pccalvo@uao.edu.co (P.C.C.); 2Clinical Investigation Research Group, School of Medicine and Health Sciences, Universidad del Rosario, Bogotá 111221, Colombia; alejandro.ondo@urosario.edu.co; 3Givia Research Group, School of Medicine and Health Sciences, Universidad del Rosario, Bogotá 111221, Colombia; dianaco.rodriguez@urosario.edu.co

**Keywords:** biosensors, carbon dots, cancer, electrochemical biosensor, glassy carbon, optical biosensor

## Abstract

Cancer remains one of the leading causes of mortality worldwide, making early detection a critical factor in improving patient outcomes and survival rates. Developing advanced biosensors is essential for achieving early detection and accurate cancer diagnosis. This review offers a comprehensive overview of the development and application of carbon dots (CDs) and glassy carbon (GC) biosensors for early cancer detection. It covers the synthesis of CDs and GC, electrode fabrication methods, and electrochemical and optical transduction principles. This review explores various biosensors, including enzymatic and non-enzymatic, and discusses key biomarkers relevant to cancer detection. It also examines characterization techniques for electrochemical and optical biosensors, such as electrochemical impedance spectroscopy, cyclic voltammetry, UV–VIS, and confocal microscopy. The findings highlight the advancements in biosensor performance, emphasizing improvements in sensitivity, selectivity, and stability, as well as underscoring the potential of integrating different transduction methods and characterization approaches to enhance early cancer detection.

## 1. Introduction

Biosensors are critical tools in analytical science since they combine biological response material with transducers to selectively measure biological or chemical analytes by converting a biological signal into a measurable electrical signal [1]. Figure 1 illustrates the elements of a typical biosensor, which consists of three fundamental components: a sensitive bioelement that recognizes the target analyte; a signal transducer that detects variations in one or more types of signals, such as impedance, electrical current, power, electromagnetic radiation, and optical density; and a signal processing component that provides readable outputs for analysis and interpretation. The analyte refers to the analyzed substance, ranging from small molecules, such as glucose and dopamine [2], to macromolecules, including proteins, nucleic acids, and polysaccharides [3]. These diverse analytes demonstrate the versatility of biosensors in addressing a wide spectrum of analytical needs across various fields, particularly in biomedical applications [4,5].

In recent years, biosensors have undergone remarkable advancements, driven by the integration of cutting-edge materials, innovative design principles, and interdisciplinary approaches. The incorporation of nanotechnology, particularly nanomaterials such as carbon dots (CDs), graphene, and metallic nanoparticles, has significantly enhanced the sensitivity, selectivity, and detection limits of biosensors. Additionally, advances in fabrication techniques, such as 3D printing and microfluidics [3,6,7], have facilitated the development of miniaturized, portable, and cost-effective devices suitable for point-of-care diagnostics. Progress in computational tools, including machine learning and artificial intelligence, has further optimized data analysis and interpretation, enabling real-time, multi-analyte detection with high accuracy [8,9]. These advancements have expanded the application scope of biosensors from traditional clinical diagnostics to personalized medicine, environmental monitoring, and food safety. In cancer research, biosensors now play a pivotal role in identifying novel biomarkers, enabling early detection, monitoring therapeutic responses, and developing targeted treatment strategies [10,11,12]. These innovations reflect the growing potential of biosensors to address contemporary challenges in healthcare and beyond, paving the way for more precise and accessible diagnostic solutions.

The biosensors are categorized based on the transduction principle and the biological material they utilize, which includes electrochemical, calorimetric, optical, piezo-electric, and resonant biosensors. Specifically, electrochemical biosensors use various techniques such as amperometric, potentiometric, voltammetric, conductometric, and impedimetric methods. These biosensors interact with the recognition elements of the biochemical analytes, generating an electrical signal to transduce the chemical response [13,14].

Electrochemical biosensors can be categorized into enzymatic and non-enzymatic sensors. Enzymatic sensors offer excellent selectivity and sensitivity at low concentrations, using enzymes like glucose oxidase, lactate oxidase, or urease as the recognition elements. However, they suffer from a rapid decrease in selectivity over time and are susceptible to external environmental factors. In contrast, non-enzymatic sensors, employing materials like conductive polymers, nanomaterials, or organic molecules instead of enzymes, have been developed to address the limitations of enzymatic biosensors. They provide long-term stability, high electron transfer rates, and high electrocatalytic activity. Nonetheless, their selectivity remains lower than that of enzymatic sensors [15,16,17].

Optical biosensors have become an increasingly popular choice for disease detection and prevention in recent decades due to their rapid response and user-friendly nature. These biosensors can be classified as either label-free or label-based [18]. Label-free optical biosensors operate without labels such as fluorescence or radio labels, offering benefits like real-time detection and avoidance of false signals caused by labeling. In contrast, label-based optical biosensors depend on labels like fluorescent dyes or nanoparticles to measure biomolecular binding events, which can sometimes interfere with the binding process and reduce efficiency [19,20]. This advancement in optical biosensors points toward a promising direction in the evolution of biosensors, particularly in cancer diagnosis and treatment, aiming to minimize unnecessary biopsies and enhance diagnostic accuracy [21,22,23]. These biosensors are being studied for the detection and treatment of skin cancers such as melanoma and non-melanoma types, and their potential application may extend to other forms of cancer, such as breast, lung, and prostate cancer, in the future [24]. Optical biosensors are designed based on optical phenomena, including fluorescence, chemiluminescence, and surface plasmon resonance (SPR) [25,26].

Nano biosensors are biosensors based on nanostructured materials that are promising tools for improving the detection of a specific analyte. CDs and GC are the most widely used nanomaterials in developing optical and electrochemical biosensors [27,28]. CDs are carbon nanoparticles with a quasi-spherical shape composed of crystalline or amorphous carbon. They primarily consist of graphene and graphene oxide sheets, whether in the form of sp^2^-graphitic carbon or from the introduction of sp^3^-hybridized carbon. They have been particularly attractive due to their nanoscale size, morphology, high colloidal stability, broad absorption in the UV-visible light spectrum, photoluminescent properties, low toxicity, and simplicity in their manufacturing processes [29,30,31]. GC is a form of disordered sp^2^ carbon, characterized by unique material properties such as high strength, a low density of approximately 1.5 g/cm^3^, the capability to withstand high temperatures in inert gas up to 3000 °C, and its exceptional extreme corrosion resistance [32,33]. These characteristics make carbon-based nanomaterials (CDs and GC) attractive for the fabrication of portable, biocompatible, and efficient nano biosensors with potential applications in fields such as cancer detection and treatment, officering improved specificity, affordability, and sensitivity compared to traditional cancer detection methodologies [34,35,36].

## 2. Development of Biosensors

Integrating nanotechnology with biosensors enables rapid and accurate detection of molecular biomarkers in different samples. Nanomaterials contribute to reducing detection limits to individual molecules and improving sensor performance by increasing the number of bioreceptor units immobilized. Organic, inorganic, metallic, and hybrid nanomaterials have emerged as key players in this field due to their unique properties and adaptability. Hybrid materials, combining metallic clusters and carbon nanostructures, are particularly effective in improving optical and electrochemical detection [37,38,39]. The synthesis of nanomaterials involves two methods classified as ‘bottom-up’ and ‘top-down’ approaches. In the ‘top-down’ approach, bulk materials are mechanically processed and converted into fine particles on the nanoscale size regime. Some techniques used are mechanical machining, physical vapor deposition (PVD), lithography, electrochemical method, and pyrolysis through thermal evaporation [40,41]. This method has the advantages of large-scale production, the possibility of deposition over a large substrate, and the fact that it does not require chemical purification. However, it also comes with disadvantages such as broad size distribution, varied particle shape, and difficulty in controlling deposition parameters, and it usually involves expensive equipment [42].

In contrast, the fine particles are assembled using a bottom-up approach to construct nanomaterials. The ‘bottom-up’ approach involves the sol–gel method, chemical vapor deposition (CVD), chemical co-precipitation, micro-emulsions, hydrothermal, and microwave methods [40,41]. This method can produce ultra-fine nanoparticles, allows for control over deposition parameters, and is generally cheaper. However, it is challenging for large-scale production and requires chemical purification [42]. The choice between these methods depends on the material and the specific applications. Applications of these methods vary widely; top-down approaches are used in areas like electronics and structural materials, while bottom-up methods are frequently employed in biomedical fields due to their ability to produce high-purity and well-defined nanomaterials [43,44].

### 2.1. Synthesis of Carbon Dots

Synthetic procedures for CDs are diverse and tailored to achieve specific characteristics and functionalities. Pyrolysis, a common technique, involves carbonizing organic precursors to yield nanoscale carbon structures with quantum confinement effects. Notably, the pyrolysis of citric acid is a prevalent approach. This method entails meticulously preparing a mixture comprising L-histidine, citric acid, and ultrapure water, which undergoes ultrasonic dissolution to ensure homogeneity. Subsequently, the solution is transferred to a Teflon-lined autoclave and heated to 180 °C for 4 h, facilitating carbonization. Centrifugation is employed to separate larger particles, followed by a dialysis procedure [45].

Another notable method is hydrothermal synthesis, which employs elevated temperature and pressure conditions within an aqueous solution to facilitate the generation of CDs from various carbon precursor molecules [46], such as plant-derived carbon precursors, specifically *Ferulago angulata*. This technique entails the carbonization of *Ferulago angulata* at different temperatures, followed by dispersion in deionized water to achieve a homogeneous solution [47]. Furthermore, CDs have been successfully synthesized from tomato juice via hydrothermal treatment, resulting in particles ranging in size from 1.3 to 3.7 nm. These tomato-derived CDs have shown utility in detecting carcinoembryonic antigens [48]. Typically, this method involves reaction conditions ranging from 90 °C to 180 °C and autogenous pressures from 10 to 40 bar in a sealed system, such as a Teflon-lined autoclave, and reaction durations of up to 12 h [49].

Microwave-assisted synthesis is a rapid and efficient technique that allows for uniform heating and precise control, significantly reducing reaction time [44]. CDs were synthesized using this technique from banana peels to manufacture a biosensor for detecting colitoxin DNA in human serum [50]. The microwave-assisted technique’s rapid and efficient synthesis of CDs doped with sulfur and nitrogen, completed within 3 min of reaction time, demonstrates a significant advancement in materials chemistry. These dots exhibit high dispersity in water and a quantum yield of fluorescence of 75.6 ± 2.1%, along with notable antimicrobial and antioxidant properties, providing an efficient and sustainable alternative to traditional methods [51]. This method typically involves heating precursor solutions at power levels of 500–800 W, with reaction times ranging from 2 to 10 min depending on the precursors and power of the microwave [49].

The electrochemical method is another viable technique for synthesizing CDs. This method involves applying a specific voltage or current to a working electrode, typically a carbon material with conductive properties, to induce an oxidation reaction at the anode. This process results in the delamination of carbon nanoparticles from the carbon source, which is then obtained as CDs after centrifugation [44]. An example of the electrochemical method’s versatility is the synthesis of nitrogen and sulfur co-doped carbon dots (N, S/CDs) through electrolysis. Electrolyzing graphite rods in a specific solution can form N, S/Graphene Dots, demonstrating the diversity of available synthesis methods [52].

### 2.2. Synthesis of Glassy Carbon

The synthesis of GC relies exclusively on pyrolysis as the fundamental method. This process involves the thermal decomposition of carbonaceous resins in an inert atmosphere (e.g., argon or nitrogen), with variations such as alternative precursors, controlled atmospheres, and tailored heating regimens to adjust material properties [53]. Among the different precursors, phenolic resins (phenol-formaldehyde) and poly (furfuryl alcohol) are the most commonly used due to their thermal stability and ability to form dense and homogeneous structures [54]. Polystyrene sulfonate has also been employed to produce uniform GC structures such as spheres. Natural precursors, like camphor, are sometimes used to simplify the pyrolysis procedure. Additionally, novel resins have emerged as alternatives to traditional thermosetting resins, opening new possibilities in the synthesis of GC [53].

Among these modifications, the synthesis of the modified glassy carbon electrode (GCE) begins with the preparation of a suspension of g-C_3_N_4_ by ultrasonically dispersing 1 mg of g-C_3_N_4_ in 4 mL of deionized water. A 2.0 μL drop of this suspension is applied to a clean GCE and dried under an infrared lamp. Subsequently, l-cysteine is electropolymerized on the g-C_3_N_4_/GCE surface via cyclic voltammetry (CV) for 240 s at a scan rate of 5 mV/s. This modified approach enhances the electrochemical properties of GC, making it practical for detecting methotrexate in pharmaceutical and biological samples [55]. Figure 2 presents a summary of the main techniques used for the synthesis of carbon-based nanomaterials, highlighting their characteristics, advantages, and recent applications. This comparison provides clear insight to guide optimal selection in future research.

### 2.3. Fabrication Methods of Electrodes

#### 2.3.1. Screen-Printed Electrodes (SPE)

SPE devices have stood out in recent decades thanks to their portability, low manufacturing cost, ease of use, and the possibility of being manufactured from different substances with flexible selectivity. Thanks to their versatility and high reproducibility, SPEs are attractive and ideal for use in various applications such as medicine, pharmacy, environment, etc. [56,57]. Cancer detection is one of the most promising areas for applying SPE [10]. These electrodes can be used in the diagnosis of various cancer-related pathologies, such as pancreatic cancer, melanomas, leukemia, breast cancer, glioma cells, cervical cancer, and ovarian cancer, to name but a few [58,59,60]. The ability of SPE to detect specific markers in biological samples makes them valuable tools for the early detection and monitoring of disease progression.

The manufacturing process of an SPE begins with the design of a mesh, which delineates the size and geometry of the electrode, as well as the selection of conductive inks and suitable substances for the substrate. The most common inks for working electrodes (WE) are carbon-based. Silver or silver chloride inks are employed for the reference electrode (RE), whereas the counter electrode (CE) typically utilizes the same ink as the WE. Subsequently, thin films are fabricated through a layer-by-layer deposition technique, applying the chosen inks onto the substrate. This approach ensures the creation of a uniform and functional layer. After applying layers, the electrodes are subjected to drying using hot air and IR radiation. A curing process is also carried out to solidify the ink, thereby enhancing the durability and stability of the electrode [57].

In the final step, the electrical circuits are coated with an insulating material. A sample is added to the surface of the SPE to conduct analytical evaluations, thus completing the manufacturing and preparation process of the electrode for use in various applications [57].

##### Screen-Printed Electrode Configurations

There are two configurations of screen-printed electrodes:-Screen-printed modules with three electrodes, which consist of a working electrode at which the electrochemical reaction of interest occurs, an auxiliary electrode (counter electrode) that completes the electrical circuit and is usually made from an inert material that does not participate in the electrochemical reaction under study, and a reference electrode that provides a stable potential against which it measures the potential of the working electrode [61].-Screen-printed modules with four electrodes, including a working electrode, a working sensor electrode, an auxiliary electrode, and a reference electrode. The four-electrode configuration is usually employed to measure the effect of an applied current on a solution or some barrier within that solution. The selection of the configuration depends on the specific application; however, the three-electrode configuration is the most used for fabricating biosensors to detect cancer [61].

It is important to note that these electrodes are available in various materials depending on the application’s needs, such as glassy carbon, gold, platinum, or silver. In this review, the focus is on the working electrode material, specifically glassy carbon.

##### Glassy Carbon Electrodes (GCEs) as Working Electrode

As mentioned above, for applications such as cell detection or cancer biomarkers, the most used material for detection or working electrodes is GC [60,61,62,63]. These types of electrodes are made from highly purified carbon and subjected to rigorous heat treatment to generate a dense, non-graphitic structure, which gives GCE its outstanding stability and durability. However, to improve their analytical properties, such as sensitivity and selectivity, and increase stability, there has been a growing interest in developing these electrodes modified with nanomaterials, especially carbon-based ones, such as graphene or CDs [64,65]. The ability of these nanomaterials to deliver enhances electrocatalytic activities and reduces deposition on the electrode surface, making them beneficial for biosensor development [61].

Modifying the working electrode with different nanomaterials can involve a few advanced techniques, the most used being electrochemical deposition and drop casting [66,67]. These strategies allow for optimizing the electrode properties and improving its detection capacity, thus contributing to developing more sensitive and selective biosensors for the early and accurate identification of cancer biomarkers.

It is essential to highlight that this modification represents just one step in modifying an electrode’s surface. Before modification, thorough preparation of the electrode surface is necessary. Typically, this involves three essential steps: polishing, cleaning, and drying, using specific materials to remove contaminants, and ensuring accurate and reproducible electrochemical measurements. A few methods are commonly used for surface polishing and cleaning. Polishing can be achieved with Al_2_O_3_ or alumina slurry powder. Both ethanol and deionized water are practical options for cleaning, and electrochemical cleaning is a well-established technique [45,68,69,70].

#### 2.3.2. Electrochemical Deposition

Electrochemical deposition, or electrodeposition or electroplating, is a method for depositing conducting or semiconducting materials onto a substrate. This process relies on the application of an electric field and redox reactions. By passing an electrical current through an electrolyte solution containing cations of the desired material, these positively charged ions are reduced at the surface of the electrode (cathode), forming a thin film or coating of the target material on the substrate [71].

Electrodeposition is a widely recognized conventional technique employed to enhance diverse materials’ aesthetic appeal and functionality by modifying their surface characteristics. However, its significance extends further as it is increasingly acknowledged as a versatile method for crafting nanomaterials. In sensor development, electrodeposition plays a pivotal role in adapting the surface properties of electrodes. This precision in surface modification is crucial for enhanced electrode conductivity; facilitating electron transfer; and improving biosensors’ analytical sensitivity, selectivity, and stability [72,73].

Electrochemical deposition is a pivotal technique in the surface modification of working electrodes for biosensor applications. In a 2021 study by Soleimanian et al., a novel electrochemical cytosensor was designed for the sensitive detection and quantification of KG1a cells as a model of LSCs. To achieve this objective, the GCE underwent modification by incorporating graphene dots (GDs), which was aimed at improving its electrical conductivity, particularly by augmenting the specific surface area of the modified electrode. This was accomplished by electrochemically depositing synthesized GDs onto the pre-cleaned GCE using the cyclic voltammetry (CV) technique [74]. The summary of studies on electrochemical deposition is shown in Table 1.

#### 2.3.3. Drop Casting

The drop casting technique is a simple, easy, and fast procedure that directly incorporates nanomaterials into the electrode. This technique is mainly used to prepare the surface of vitreous and silk-screened carbon electrodes [75]. In this method, a drop of liquid containing a suspension of the particles of the nanomaterial of interest is deposited directly on the surface of the electrode to be modified with the help of a suitable solvent, ideally limiting its distribution exclusively to the working electrode without spilling onto the surrounding insulating material [76].

In a 2019 study by Guangxi Normal University, the electrode modification process primarily utilized the drop casting technique. A total of 6 μL of primary anti-CA15-3 (Ab1) solution (50 μg mL^−1^) was initially deposited onto the PDA-AgNPs (nanocomposites of Ag nanoparticles and polydopamine)/GCE electrode. This deposition occurred at 4 °C over 12 h, forming Ab1/PDA-AgNPs/GCE. Subsequently, to eliminate physically adsorbed Ab1, the electrode was rinsed with PBS. Next, 4 μL of bovine serum albumin solution (BSA) 0.5 wt% was drop-cast onto the modified electrode and incubated at 37 °C for 40 min. This step effectively blocked nonspecific coupling sites, yielding BSA/Ab1/PDA-AgNPs/GCE. The drop casting technique facilitated precise control over the deposition of each layer, ensuring efficient modification of the electrode surface [45].

In a study conducted in 2021 by Doong et al., the biosensing electrode was crafted through drop-casting the nanocomposite onto the GCE. Specifically, the sulfur-doped graphene quantum dots (S-GQDs) and gold-decorate carbon nanosphere (Au-CN) nanocomposite (S-GQDs@Au-CNs) was deposited onto the polished GCE electrode to form the sensing probe. This process yielded the GCE||Au–CNS@S-GQD/Ang-2 configuration [77].

In a 2022 study by the Academy of Scientific and Innovative Research, the drop casting technique was employed to prepare the biosensing electrode. Initially, 5 μL of GQD solution was drop-cast onto the GCE surface and allowed to dry at ambient temperature for 24 h. Subsequently, a freshly prepared solution of 4:1 EDC: NHS in 10 mM PBS (pH 7.0) was drop-cast onto the GQD-modified electrodes to activate the functional groups. After one hour, the electrodes were rinsed with Milli-Q water. Next, 5 μL of 20 μg/mL CD44 antibodies (in 10 mM PBS pH 7.0) was immobilized to allow the antibodies to bind with the activated electrode surface. The final surface-modified GCE, i.e., BSA/CD44 antibody/GQDs/GCE, was used as the electrochemical probe to detect the CD44 antigen [78].

In a recent 2023 study by Jilin University, SnS_2_ nanosheets were applied onto a GCE, followed by the deposition of DSPE-PEG_2000_-NH_2_ solution onto the same electrode, which was then subjected to incubation. Subsequently, DOPC solution was added and incubated, resulting in the formation of a fluid layer. Next, 4 μL of capture DNA (H1) modified with cholesterol was incubated on the surface of the lipid bilayer at room temperature. During the sensing process, different concentrations of target miRNA-27a-3p were introduced and incubated at room temperature. Finally, Mo_2_TiC_2_ QDs (bimetallic MXene derivative)-labeled H_2_ was introduced into the sensing system at room temperature and allowed to incubate. After removing any unconnected miRNA and Mo_2_TiC_2_ QDs, an ECL test was conducted in phosphate-buffered saline containing potassium persulfate (K_2_S_2_O_8_) [79]. The summary of the information about these studies and other relevant ones is found in Table 1. micromachines-16-00139-t001_Table 1Table 1Resume of drop casting and electrochemical deposition studies.Electrode and Modification *Modification TechniqueYearRef.Au NCs/MWCNTs-NH**_2_**/Ab**_2_**Drop cast2018[80]**BSA/Ab1/PDA-AgNPs/GCE**Drop cast2019[45]**GCE/NHCDs/CS/Au NPs/Con A**Both2020[52]GCE||Au–CNS@S-GQD/Ang-2 Drop cast2021[77]ssDNA/Cys–ZnS-QD/GCEElectrodeposition2021[81]Ab/GCE/GQD/AuNPs/St@AuNPsElectrodeposition2021[74]BSA/CD44 antibody/GQDs/GCEDrop cast2022[78]GCE/CoP-BNF/SNGQDs@AuNPs/TrasmatuzabDrop cast2022[68]AuNPs–WS**_2_**QDs–GCEBoth2022[82]GCE/SnS**_2_** nanosheets/lipidbilayer/Mo**_2_**TiC**_2_** QDs—GCE/lipid bilayer/Mo**_2_**TiC**_2_** QDsDrop cast2023[79]GCE-OLC; GCE-OLC-PAN; GCE-OLC-PANDrop cast2023[83]* Au NCS, Au nanocages; MWCNTs-NH_2_, amino-functionalized multiwalled carbon nanotubes; Ab2, breast cancer ER antibody; Conv A, concanavalin A; Ang 2, angiopep-2; SsDNA, NH_2_-probe; Cys-Zns-QD, L-cysteine ZnS quantum dots; CoP-BNF, gold nanoparticles and a porphyrin binuclear; SNGQDs, sulfur/nitrogen-doped GQDs; OLC-PAN, onion-like carbon and polyacrylonitrile composites.


#### 2.3.4. Electrodeposition and Drop Cast

Researchers have used electrodeposition and drop casting techniques in recent studies to modify GCE surfaces for various electrochemical applications. These techniques offer distinct advantages and enhanced functionalities and, when combined, improve the performance of the modified electrodes. 

In a 2020 study by Southwest University, a mixture solution containing nitrogen-doped carbon dots (NHCDs) and chitosan (CS) was drop-cast onto a well-polished GCE surface. After drying, gold nanoparticles (AuNPs) were electrodeposited onto the NHCDs/CS film [52]. 

Another study in 2022 by Pourakbari et al. involved the sequential modification of the GCE surface by drop casting tungsten disulfide quantum dots (WS_2_QDs) followed by electrodeposition of AuNPs using the cyclic voltammetry (CV) technique [82].

### 2.4. Transducer Principles

#### 2.4.1. Electrochemical Principles of Transduction

##### Potentiometric Biosensors

Potentiometric biosensors generate a potential difference between the working electrode and the reference electrode to detect chemical reactions of electroactive materials under a constant current. This technique does not deplete the measured species compared to amperometric biosensors. The value of potentiometric biosensors lies in their sensitivity and selectivity, particularly when coupled with a reliable reference electrode, which contributes to their significance in various applications. Moreover, their manufacturing simplicity and performance maintenance, even with significant size reduction, further enhance their appeal [41,84]. Despite their structural similarities, a potentiometric biosensor was implemented using an aptamer-based nano filter interface to distinguish between l-3,4 dihydroxyphenylalanine (l-DOPA) and dopamine (DA). A DA–Ap nano_filter-coated Gold (Au) gate field-effect transistor (FET) was employed, with an Au gate electrode assessing the electrical response to l-DOPA based on oxidative reactions. The nano_filter interface, formed by immobilizing the DA–Ap layer on an aryl-diazonium-based anchor monolayer, allowed for clear discrimination between the electrical signals of l-DOPA and DA [85].

##### Amperometric Biosensors

Amperometric biosensors stand out for their ability to generate current via oxidation reactions, making a device capable of providing precise quantitative analytical data. While these biosensors often compete with potentiometric biosensors in response time, energy range, and sensitivity, they may face challenges regarding selectivity and susceptibility to interference from other electroactive substances. These biosensors operate in either two-electrode or three-electrode configurations. They measure the current produced by electrochemical oxidation or reduction of electroactive species at the working electrode. This occurs when a constant potential is applied with respect to the reference electrode. When a potential is applied during operation, the resulting current (typically from nanoamps to milliamps) reflects the catalytic conversion or protein adsorption occurring at the electrode surface in the amperometric biosensor setup [41,84]. An amperometric biosensor for l-fucose detection was developed to address the challenges of electroactive interferences in urinalysis, such as ascorbic acid, dopamine, and uric acid. By utilizing the direct electron transfer type bioelectrocatalysis of pyrroloquinoline quinone (PQQ)-dependent pyranose dehydrogenase from Coprinopsis cinerea (CcPDH), the biosensor achieved a catalytic current at a lower potential than the interfering compounds, allowing for the selective detection of l-fucose [86].

##### Conductometric Biosensors

Conductometric biosensors serve as pivotal tools for quantifying alterations in conductance between electrodes, reflecting changes in the conductivity properties of the analyte due to electrochemical reactions. These biosensors find extensive application in monitoring metabolic processes within living biological systems, facilitating real-time assessment of biochemical changes based on conductivity variations. Their utility lies in providing valuable insights into dynamic metabolic activities, enabling a deeper understanding of biological processes at a molecular level. One of the key applications of conductometric biosensors is quantifying the change in the electrical conductivity of cell solutions. Electrochemical reactions within the solution produce electrons or ions, altering the solution’s conductivity. Although the sensitivity of conductance measurement is relatively low, applying a sinusoidal electric field can mitigate undesirable effects such as Faradaic processes, concentration polarization, and double-layer charging [41,87].

##### Impedimetric Biosensors

Impedimetric biosensors detect alterations in electrical impedance at the electrode/electrolyte interface upon applying a small sinusoidal excitation signal. This method involves applying a low-amplitude AC voltage to the sensor electrode and measuring in-phase and out-of-phase current responses across different frequencies facilitated by an impedance analyzer. Such an approach enables sensor performance evaluation across a range of frequencies, offering valuable insights into analyte activity and facilitating precise measurements and monitoring. These label-free techniques are beneficial for quantifying biomolecular interactions, including enzymatic reactions, DNA hybridization, various antigen–antibody interactions, and protein–protein interactions. When a target biomolecule interacts with a specific bioreceptor on the sensor surface, changes in dielectric constant or resistance occur exclusively due to the presence of the target molecules. Consequently, impedance sensing eliminates the need for labels, making it advantageous for protein detection and constructing third-generation biosensors [41,84]. A novel four-electrode-based impedimetric biosensors were developed to assess tamoxifen’s cytotoxicity on cervical cancer cell lines, particularly HeLa cells. By employing the electric cell-substrate impedance sensing (ECIS) method, the biosensors measured cell impedance across a frequency range of 100 Hz to 1 MHz. The results indicated a significant reduction in the number of HeLa cells on the electrode surfaces in a dose-dependent manner upon exposure to tamoxifen [88].

Electro-bioimpedance is a non-invasive detection method that measures the electrical impedance of biological tissue, similar to microfluidic cell detection. It is widely used for medical purposes, complementing physicochemical and biochemical techniques. The electrical impedance (Z) is expressed as a complex relationship between the voltage response (Vo) and the current excitation (Io) flowing through the material, where ω is the angular frequency, and ϴ is the signal phase. The complex number Z is represented by the real part of the impedance (R_Z_), the resistance, and the imaginary part (X_Z_), the reactance. In a biological system, the parameter R_Z_ can be associated with the degree of hydration. The parameter X_Z_ is associated with the capacitive behavior of cells and can be compared to the number of cells present and the reactance value, and ϴ is considered a parameter that describes tissue behavior and, according to some authors, the diagnosis of clinical parameters [89].

##### Voltammetric Biosensors

Voltammetry, an electro-analytical method, quantifies current changes consequent to voltage fluctuations. Various techniques, including differential pulse voltammetry (DPV), cyclic voltammetry (CV), linear sweep voltammetry (LSV), and square wave voltammetry (SWV), among others, find application within this domain. The inherent advantages of voltammetric methods, such as their cost-effectiveness, remarkable selectivity, and heightened sensitivity, render them prevalent in biosensing systems [84]. An example of this is using the voltammetric principle to investigate the binding of pazopanib with dsDNA using bare and modified GCE. The interaction was primarily evaluated based on the decrease in the voltammetric signal of deoxyadenosine by differential pulse voltammetry. The study employed three methods for this evaluation: incubated solutions, a dsDNA biosensor, and a nanosensor. The nanosensor was fabricated using SnO_2_ nanoparticles and a carbon hybrid material derived from waste masks, the most used personal protective equipment during the COVID-19 pandemic. The results indicated that pazopanib (PZB) was active in the minor groove region of DNA [90].

#### 2.4.2. Optical Principles of Transduction

##### Fluorescence-Based Optical Biosensors

Fluorescence, an optical phenomenon utilized for analyte or molecule detection, has become a focal point in developing fluorescence-based optical biosensors. These biosensors, renowned for their exceptional selectivity, sensitivity, and rapid response time, are extensively explored in medical diagnosis, environmental monitoring, and food quality assessment [91,92,93]. They employ various fluorescent materials, such as quantum dots, organic dyes, and fluorescent proteins, enabling detection across a wide range of analytes [41]. Three primary approaches characterize fluorescence-based biosensors: fluorescent quenching (turn-off), fluorescent enhancement (turn-on), and fluorescence resonance energy transfer (FRET). These latter have gained prominence for their heightened sensitivity, particularly in studying intracellular processes. FRET involves nonradiative energy transfer from an excited donor molecule (D) to an acceptor molecule (A) at the ground state, facilitated by long-range multipole interactions. Due to their ability to detect minute changes ranging from angstroms to nanometers, FRET-based sensors find critical applications in cancer therapy and aptamer analysis. Fluorescence typically arises from the emission of light or radiation upon external exposure to an object absorbing light or radiation. A fluorescence detection system comprises four essential components: an excitation light source, a fluorophore, wavelength filters isolating emission photons from excitation photons, and a detector [84,94,95]. A novel switch-conversational radiometric fluorescence biosensor (SCRF biosensor) for highly sensitive miRNA detection was designed. This biosensor employed a structure-convertible DNA switch, a single-strand DNA with a stem-loop structure modified with two fluorophores (Cy3 and Cy5), and a quencher at specific sites. The detection process involved the production of amplicon fragments (c*) through an exponential amplification reaction. When these c* fragments hybridized to the loop of the DNA switch, the switch’s structure converted, leading to fluorescence resonance energy transfer between Cy5 and Cy3. This transfer resulted in the observation of two fluorescence signals with different trends. By analyzing the ratio of these two signals, the target miRNA could be quantitatively and rapidly detected within a concentration range from 100 fM to 100 nM, with an impressive detection limit down to 70.9 fM [96].

##### Chemiluminescence-Based Optical Biosensors

Chemiluminescence, the phenomenon wherein light energy is emitted from a chemical reaction, has garnered significant interest due to its simplicity, low detection limit, wide calibration range, and cost-effective instrumentation. Chemiluminescence-based biosensors have emerged as valuable tools in various fields. Recent advancements in chemiluminescence studies involve the integration of nanomaterials to enhance intrinsic sensitivity and explore novel detection applications. Chemiluminescence shares similarities with fluorescence, yet a crucial distinction lies in its initiation mechanism. While fluorescence relies on exciting molecules with external light, chemiluminescence triggers biomolecule excitation through chemical reactions, typically involving oxidizing substances like O_2_ or H_2_O_2_. Notably, chemiluminescence does not necessitate an external light source to initiate the reaction [41,84,94,97]. The enzyme-free chemiluminescence immunoassay with ODI-CL detection has been developed for the early diagnosis of thyroid cancer. This method involves a sandwich immunoassay using fluorescent microsphere-conjugated detection antibodies and paramagnetic beads to capture and detect thyroid-stimulating hormone (TSH) in human serum. The assay has a dynamic range of 0.037–18 μIUmL^−1^ with a low detection limit of 0.011 μIUmL^−1^, showing statistically acceptable accuracy, precision, and reproducibility [98].

##### Surface-Plasmon-Resonance-Based Biosensors

Surface plasmon resonance (SPR)-based biosensors utilize surface plasmon waves to detect changes in refractive index resulting from molecular interactions at a metal surface. This label-free biosensing technology operates on the principle of SPR, wherein polarized light illuminates a metal surface at the interface of two media with different refractive indices, generating electron charge density waves known as plasmons. The intensity of reflected light decreases at a specific angle, known as the resonance angle, proportional to the mass on the surface. Noble metals like gold and silver are commonly used to create surface plasmons in SPR biosensors. In SPR, incident light at the resonance angle excites surface plasmons, causing a decrease in reflected light intensity. Any change on the metal surface, such as biomolecule binding, alters the resonance angle, known as the SPR shift, which is proportional to the change in surface mass. A typical SPR biosensor comprises a gold surface functionalized with bioreceptor molecules for target molecule capture. Real-time detection is achieved by monitoring changes in optical reflectivity as target molecules bind to the surface [41,84,94,99]. Researchers have developed a 1-D grating-based SPR biosensor and a carboxyl-functionalized molybdenum disulfide SPR detection assay, focusing on the detection of epidermal growth factor receptor (EGFR) and carcinoembryonic antigen (CEA) lung cancer biomarkers. The biosensor’s design includes periodic gratings and multiple metal layers to enhance resonance effects. Using finite difference time domain simulations, the study confirms the biosensor’s ability to detect biomarker-induced refractive index changes, enabling label-free early lung cancer detection [100].

##### Optical-Fiber-Based Biosensors

Optical fiber biosensors, also known as bio-optrodes, are sensor systems derived from optical fibers that utilize optical fields to detect and quantify biological species such as whole cells, proteins, and aptamers. These biosensors offer a promising alternative to traditional biomolecule assessment methods. One dependable optical fiber technique involves evanescent field sensing, mainly observed in tapered optical fibers. An evanescent wave is generated at the sample interface when light passes through an optical fiber due to total internal reflection. This field decays exponentially with distance from the interface, and it can be utilized to excite fluorescence near a sensing surface. Tapered optical fibers are commonly employed with various optical transduction processes, including variations in refractive index, absorption, fluorescence, and surface plasmon resonance (SPR). Optical fibers are typically silica or plastic, characterized by their small diameter, strength, durability, and flexibility. These fibers can withstand harsh and hazardous environments, making them suitable for remote sensing applications. Additionally, optical fibers enable multiplex sensing by transmitting multiple light signals simultaneously. They consist of a cylindrical core surrounded by cladding and function as circular waveguides [41,84,94,101]. A novel lasso-shaped fiber laser biosensor was developed for specific detection of carcinoembryonic antigen (CEA)-related cell adhesion molecules 5 (CEACAM5) protein in serum samples. Emphasizing the need for high sensitivity and reproducibility in cancer biomarker detection for point-of-care testing (POCT), the biosensor offers a solution by capitalizing on the optical-fiber-based transduction principle. The biosensor operates based on changes in the spectral characteristics of a fiber laser induced by biomolecular binding. The ultra-narrow linewidth of the laser facilitates precise spectral analysis, enabling the detection of minute variations in the lasing signal resulting from biomolecular interactions [102]. Figure 3 provides an overview of each optical biosensor type’s fundamental principles and configurations, highlighting their unique mechanisms for detecting and quantifying biological species.

## 3. Types of Biosensors for Cancer Detection

### 3.1. Enzymatic and Non-Enzymatic Biosensors

Enzymatic biosensors use enzymes as the recognition element to detect and measure specific substances in a sample [103]. These biosensors are designed to convert a biochemical signal from an enzymatic reaction into a measurable output, often an electrical signal. This type is characterized by its high specificity for their target molecules, which is also related to selectivity to detect and measure the concentration of a particular analyte, reducing the interference from other substances present in the sample [104]. It also can often detect low concentrations of the target analyte, providing a sensitive response to changes in concentration. The most relevant fact is that enzymatic biosensors are typically biocompatible, allowing their integration into biological systems without causing any harm [105].

It is also necessary to identify that enzymatic biosensors have some limitations, such as their sensibility to environmental factors such as pH or temperature fluctuations. The immobilized enzymes may degrade over time, affecting the biosensor’s performance. This can introduce a new research question about how enzymatic biosensors can be improved to reduce their possible difficulties [106].

Non-enzymatic biosensors are an alternative approach that aims to overcome some limitations by using non-biological recognition elements while maintaining high sensitivity and selectivity. To better understand the functioning of this kind of biosensor, it is essential to highlight that they can detect and measure specific substances in a sample without relying on enzymes as the recognition element. Instead, they use non-biological materials or components to achieve selective and sensitive detection of target analytes. Non-enzymatic biosensors can overcome the limitations of enzymatic biosensors by exhibiting stable long-term characteristics, high-cost performance, high sensitivity, electron transfer, and high electrocatalytic activity [107]. Also, their resistance to environmental factors makes them more robust in diverse operating conditions, and they can be manufactured using a wider variety of materials compared to the enzymatic ones [5].

The limit of detection (LOD) is a critical parameter in biosensor performance, representing the lowest concentration of an analyte that can be reliably distinguished from the absence of the analyte (blank sample). It is influenced by factors such as sensor sensitivity, noise levels, and the specificity of the biorecognition element [108]. In the case of enzymatic biosensors, LOD values typically range from nanomolar (nM) to micromolar (µM) levels, depending on the enzyme’s catalytic efficiency and the detection method employed. On the other hand, non-enzymatic biosensors, which rely on direct physicochemical interactions rather than biological elements, often achieve lower LOD values, extending into the picomolar (pM) range, due to their enhanced stability, reproducibility, and the potential for miniaturization [104,109]. The selection of an enzymatic or non-enzymatic biosensor depends largely on the desired sensitivity, operational stability, and the specific application in clinical or environmental monitoring.

Enzymatic biosensors hold significant promise for cancer applications due to their ability to detect specific biomolecules associated with cancer development and progression. These biosensors typically consist of enzymes integrated with transducers that convert biochemical signals into measurable electrical or optical outputs [68,79]. In the context of cancer, enzymatic biosensors can detect cancer-specific biomarkers such as enzymes, proteins, or nucleic acids in bodily fluids or tissue samples with high sensitivity and specificity. This enables early cancer diagnosis, disease progression monitoring, and treatment efficacy assessment. Moreover, enzymatic biosensors offer rapid detection, low cost, and portability, making them suitable for point-of-care testing and remote monitoring applications.

Non-enzymatic biosensors represent a promising avenue for cancer applications, offering distinct advantages over enzymatic counterparts. These biosensors utilize non-biological recognition elements such as aptamers, antibodies, or molecularly imprinted polymers to bind to cancer-specific biomarkers selectively. By leveraging the unique molecular interactions between these recognition elements and target molecules, non-enzymatic biosensors can detect cancer biomarkers with high specificity and sensitivity [110]. Additionally, their compatibility with various transduction mechanisms, including electrochemical, optical, and surface plasmon resonance, enables versatile detection platforms tailored to specific cancer biomarkers and sample types.

### 3.2. Biomarkers

Biomarkers are measurable indicators of biological processes, conditions, or states within an organism. These can be objectively measured substances, molecules, or genes, aiding disease diagnosis, treatment selection, and monitoring therapeutic efficacy [111].

Biomarkers in cancer provide valuable insights into the development, progression, and response to the treatment of cancerous tumors. Biomarkers such as specific genetic mutations, protein expression levels, or abnormal cellular features serve as diagnostic tools to identify and characterize the existence of cancer [112]. Additionally, it is possible to use biomarkers to recognize the prognostic of cancer patients, guiding clinicians in determining the most appropriate treatment strategies and personalized therapies. Through the monitoring of biomarkers during treatment, healthcare providers can assess the effectiveness of therapies and make informed decisions regarding adjustments or changes to treatment regimens. Moreover, biomarker-driven clinical trials enable the development of targeted therapies tailored to individual patients, ultimately improving outcomes and advancing the field of oncology [113].

For the case of skin cancer, and acknowledging the existent types of this, it is possible to identify biomarkers related explicitly to the presence of this condition. The most common biomarkers associated with skin cancer encompass a range of molecules and proteins reflective of disease progression and aggressiveness. In melanoma, multiple biomarkers are identified as diagnostic tools, such as Human Melanoma Black-45 (HMB-45), Melan-A, tyrosinase, microphthalmia transcription factor, and S100, which are described as immunohistochemical markers because they allow the detection of this type of cancer despite its cytomorphological variants [114]. When checking on the most common types of skin cancer, it is essential to introduce two kinds of carcinoma in this section: basal cells (BCCs) and cutaneous squamous cells (CSCCs). BCCs have a characteristic related to the difficulty they present with differentiation from trichoblastoma, so there are two novel biomarker candidates: Meteorine-Like Peptide (METRNL) and Asprosin, where METRNL presented an overexpression in the lesion area of the trichoblastoma, but Asprosin did not increase, but in BCC samples, they both are relatively higher [115]. Otherwise, CSCCs are vitally detected through MYBL2 and TK1, demonstrating the cancer progression in analyzing differentially expressed gene (DEG) samples [116].

The articles reviewed in the paper’s research phase identified different biomarkers related exclusively to skin cancer using biosensors of various types and detection mechanisms with their operation conditions. It is also important to mention the materials used to build up the biosensor to clarify the features related to their properties. In 2019, CDs and molecular-beacon-based and AuPt nanoparticles, vertical graphene (VG) sheets, and GCE were used for the manufacturing of biosensors that detected MicroRNA-21 and alpha-fetoprotein (AFP), respectively. They are both skin cancer biomarkers. The CDs and molecular beacon-based one detect MicroRNA-21. In contrast, AuPt nanoparticles, vertical graphene (VG) sheets, and GCE are used to detect alpha-fetoprotein (AFP). These biosensors demonstrate the potential of using advanced materials and detection mechanisms for the early and accurate diagnosis of skin cancer.

Then, B16-F10 cells were detected as skin cancer biomarkers. Still, there is particularity in its classification of biosensors because they do not fit in any of the analytes related to the enzymatic or non-enzymatic structure; thus, they can be classified under a different differentiation criterion. Finally, in 2023, using antimonide nano-flakes (AMNFs) and CDs, it was possible to sense MicroRNA-21 again using an electrochemical detection mechanism. It also included the detection of HER2 (human epidermal growth factor) with a biosensor made with carboxylic-acid-group-rich graphene quantum dots (GQDs) modified with gold nanoparticles and a porphyrin binuclear framework (CoP-BNF) to modify the GCE, because it is noted that some biomarkers are not specific for just one type of cancer but for grouping them according to the type of cells or tissues involved [68]. A summary of the biomarkers utilized in various biosensors and the respective detection conditions can be found in Table 2.

## 4. Characterization of Biosensors

### 4.1. Characteristics of a Biosensor

The primary attributes of biosensors are selectivity, linearity, stability, repeatability, and sensitivity. Specific analytes can be identified among other chemicals according to selectivity, and consistent results from repeated experiments are guaranteed thanks to repeatability. Stability refers to the biosensor’s resistance to environmental perturbations to provide accurate measurements throughout time. The ability to detect minute amounts of an analyte is determined by sensitivity, which is important for medical applications. Finally, linearity guarantees a precise and direct correlation between the measured response and the analyte concentration, necessary for accuracy over a range of concentrations [105,118,119,120,121].

### 4.2. Techniques for the Characterization of Electrochemical Biosensors

#### 4.2.1. Electrochemical Impedance Spectroscopy (EIS)

EIS is a technique involving the application of an alternating current signal to an electrochemical system and measuring the frequency response of the resulting impedance. It allows the characterization of resistance and capacitance at the electrode–electrolyte interface and charge transfer processes and chemical reactions at the interface [121,122]. Electrochemical impedance spectroscopy is a technique involving the application of an alternating current signal to an electrochemical system and measuring the frequency response of the resulting impedance. It allows the characterization of resistance and capacitance at the electrode–electrolyte interface, as well as charge transfer processes and chemical reactions at the interface [121,122].

EIS has emerged as a crucial tool in detecting and analyzing various biomarkers and biological entities. In a 2021 study, EIS was employed to develop an impedimetric detection system for glioma cells using sulfur-doped GQDs and gold-carbon nanospheres. This system proved effective in buffered solutions and complex biological samples like human serum, highlighting its potential for real-time biomedical applications [77]. Furthermore, in a subsequent study from 2022, EIS was utilized to confirm the successful immobilization of DNA probes on CDs, vital for susceptible detection of the BRCA1 gene in real samples and cellular imaging. The stabilization of DNA probes on the surface of quantum dots was verified through fluorescence spectroscopy and EIS [47].

Additionally, in an innovative approach in 2023, a bimetallic MXene quantum-dot-based electrochemiluminescence (ECL) sensor was developed for miRNA-27a-3p detection. EIS played a crucial role in confirming the successful construction of the sensor system, and the linear relationship between ECL intensity and miRNA-27a-3p concentration demonstrated the biosensor’s selectivity and sensitivity [79]. These studies underscore the fundamental role of EIS in developing and characterizing biomolecular detection systems, paving the way for research and application in clinical diagnostics and biotechnology. The characteristics of the studies that used EIS for electrochemical characterization are found in Table 3.

#### 4.2.2. Cyclic Voltammetry (CV) 

CV is an electrochemical technique where a reversal experiment is conducted by changing the direction of the potential scan at a specific time or at a designated switching potential. During the experiment, the applied potential varies linearly with time, and upon reaching the reversal point, the scan direction is reversed. This is done to observe the electrochemical response of the system in both scan directions [121,122,123,124]. Cyclic voltammetry is an electrochemical technique where a reversal experiment is conducted by changing the direction of the potential scan at a specific time or at a designated switching potential. During the experiment, the applied potential varies linearly with time, and upon reaching the reversal point, the scan direction is reversed. This is done to observe the electrochemical response of the system in both scan directions [121,122,123,124].

CV is a pivotal technique in the electrochemical characterization of biosensors, as demonstrated in two distinct studies. In the first investigation from 2019, a novel electrochemiluminescence (ECL) immunosensor for detecting CA15-3 in human serum was meticulously evaluated. The sensor exhibited remarkable stability through continuous cyclic voltammetry, as evidenced by consistent ECL signals across multiple scanning cycles [45].

The ECL immunosensor demonstrated high selectivity, effectively distinguishing interference substances and explicitly responding to samples like CA15-3. It exhibited remarkable reproducibility, with cyclic voltammetry curves remaining nearly constant over 11 consecutive scans and an RSD below 2.3% for CA15-3 measurements. Additionally, it showed stability with only a 10.4% decrease in ECL signal after one month of storage at 4 °C. The sensor achieved a LOD of 0.0017 U mL^−1^ and maintained a linear concentration range from 0.005 to 500 U mL^−1^ [34].

In a complementary study from 2020, catalase-immobilized antimonide quantum dots (Cat@AMQDs) were explored as an electrochemical biosensor for the quantitative determination of H_2_O_2_ from CA-125 diagnosed ovarian cancer samples. Through CV, the modified electrode exhibited distinct redox activity, featuring oxidation and reduction peaks indicative of the redox activity of Cat@AMQDs-GCE [58]. Together, these studies underscore the utility of CV in the comprehensive electrochemical characterization of biosensors, elucidating their stability, selectivity, reproducibility, and optimal operating conditions for diverse clinical applications.

The Cat@AMQDs-GCE electrode demonstrated selectivity for H_2_O_2_ detection, even in interferences like ascorbic acid and glucose. The modified electrode retained its redox behavior over 30 cycles, with a recovery rate of 95% to 103.4%. Additionally, the electrode’s stability was evaluated through 30 cyclic voltammetry cycles for 1 mM H_2_O_2_ in 0.1 M PBS at pH 7. The LOD was 4.4 μM, making it suitable for biological analysis, and it showed linearity up to 0.989 [47].

Together, these studies underscore the utility of CV in the comprehensive electrochemical characterization of biosensors, elucidating their stability, selectivity, reproducibility, and optimal operating conditions for diverse clinical applications.

In recent years, significant progress has been made in developing biosensors for detecting cancer cells, utilizing electrochemical techniques such as CV and EIS. In a study conducted in 2018, an impedimetric biosensor was employed to identify cancer cells using the carbohydrate-binding ability of Concanavalin A (ConA). CV and EIS tests were performed on an Au||ConA-GQD@Fe_3_O_4_ electrode to characterize its performance. CV demonstrated a decrease in the Fe(CN)_6_^3−/4−^ peak intensity with increasing glucose concentration, demonstrating the immobilized ConA’s ability for glucose detection. At the same time, EIS revealed changes in charge transfer resistance (Rct) upon adhering to different cell lines, notably improving impedance after incubation with cancerous cells HeLa and MCF-7 [125]. In another study from 2019, a strategy based on the concatenation of aptamers–DNA and quantum dots was proposed for the ultrasensitive detection of tumor cells through mercury-free anodic stripping voltammetry. Both EIS and CV were employed to investigate the biosensor assembly process, unveiling alterations in electronic transfer resistance (Ret) and inhibiting the electron transfer process by adding biomolecules [62]. In recent years, significant progress has been made in developing biosensors for detecting cancer cells, utilizing electrochemical techniques such as CV EIS. In a study conducted in 2018, an impedimetric biosensor was employed to identify cancer cells using the carbohydrate-binding ability of Concanavalin A (ConA). CV and EIS tests were performed on an Au||ConA-GQD@Fe_3_O_4_ electrode to characterize its performance. CV demonstrated a decrease in the Fe(CN)_6_^3−/4−^ peak intensity with increasing glucose concentration, demonstrating the immobilized ConA’s ability for glucose detection. At the same time, EIS revealed changes in charge transfer resistance (Rct) upon adhering to different cell lines, notably improving impedance after incubation with cancerous cells HeLa and MCF-7 [125]. In another study from 2019, a strategy based on the concatenation of aptamers–DNA and quantum dots was proposed for the ultrasensitive detection of tumor cells through mercury-free anodic stripping voltammetry. Both EIS and CV were employed to investigate the biosensor assembly process, unveiling alterations in electronic transfer resistance (Ret) and inhibiting the electron transfer process by adding biomolecules [62].

In the same year, an immunosensor for quantitatively detecting the breast cancer biomarker UBE2C was also developed. CV and EIS tests during immunosensor fabrication showed significant changes in charge transfer resistance (Rct), reflecting modification of the electrode/electrolyte interface and obstruction of the electron transfer process due to biomolecule immobilization [69]. Furthermore, during that period, a polyaniline-decorated GQDs nanowire was suggested for impedimetric detection of the carcinoembryonic antigen (CEA). CV and EIS tests revealed alterations in current intensity and charge transfer resistance upon the electrode surface modification with biomolecules, underscoring the significance of GQDs in amplifying electrode functionality and achieving effective CEA detection [126].

A study conducted in 2021 delved into the utilization of an integrated 0D/2D heterostructure comprising bimetallic CoCu-ZIF nanosheets and CDs derived from MXene for impedimetric cytodection of B16-F10 melanoma cells. The electrochemical measurements involved various techniques, including EIS and CV, to characterize cytosensor fabrication and cell detection [117]. Similarly, during the same year, another investigation introduced an impedimetric aptasensor for the HER2 biomarker utilizing GQDs, polypyrrole, and electrodes modified with cobalt phthalocyanine. Cyclic voltammetry studies provided insight into various modified electrodes’ electron transfer properties. Additionally, EIS yielded valuable data to characterize the interface properties of the modified electrodes, assisting in their differentiation and characterization [64].

In a study conducted in 2022, the CV and EIS measurements were performed on various modified electrodes in 1 mM [Fe(CN)]_6_^3−/4−^ (in 0.1 M KCl) electrolyte. The CVs exhibited significant changes in peak potentials and shapes upon modification, indicating alterations in electron transfer properties. Meanwhile, the EIS responses, represented by Nyquist plots, provided insights into the electrode surface’s charge transfer resistance (Rct). The Rct values were used to assess the conductivity of the modified surfaces, with higher Rct indicating hindered electron transfer [68]. In addition, in a study from 2023, cyclic voltammetry was employed to investigate electron transfer properties among modified surfaces using ferricyanide as a marker. The ΔEp values, representing anodic to cathodic peak potential separation, were analyzed to evaluate electron-transporting abilities. Lower ΔEp values were indicative of desirable electron transfer properties. EIS was also used to study surface-modified electrodes, with Rct values obtained to characterize the charge transfer resistance. The data from both CV and EIS agreed, with surfaces exhibiting lower ΔEp values demonstrating lower Rct values, suggesting enhanced conductivity [70]. These investigations highlight the crucial role of cyclic voltammetry and electrochemical impedance spectroscopy in advancing biosensor technology, showcasing their versatility and effectiveness in electrochemical analysis and sensor development. The characteristics of studies that use EIS and CV for electrochemical characterization are found in Table 4.

#### 4.2.3. Differential Pulse Voltammetry (DPV) 

DPV is an electrochemical technique that allows for even better sensitivities than normal pulse voltammetry. This technique is based on a scheme of reduced-amplitude pulses, where the base potential is steadily changed in small increments for most of a drop’s lifetime. The pulse height is constant relative to the base potential, and two current samples are taken during each pulse cycle. The difference in current between these two samples is recorded against the base potential, which characterizes the electrochemical response of the system. Differential pulse voltammetry is used to study complex electrochemical reactions and is especially useful for detecting analytes with high sensitivity and selectivity [121,123]. Differential pulse voltammetry is an electrochemical technique that allows for even better sensitivities than normal pulse voltammetry. This technique is based on a scheme of reduced-amplitude pulses, where the base potential is steadily changed in small increments for most of a drop’s lifetime. The pulse height is constant relative to the base potential, and two current samples are taken during each pulse cycle. The difference in current between these two samples is recorded against the base potential, which characterizes the electrochemical response of the system. Differential pulse voltammetry is used to study complex electrochemical reactions and is especially useful for detecting analytes with high sensitivity and selectivity [121,123].

In a 2019 study, EIS was utilized to analyze the interfacial properties of electrodes throughout various modification stages, from the GCE to the final sensor formation. Nyquist plots revealed changes in the charge transfer resistance (Rct), confirming the success of immunological reactions at each electrode modification step. Differential pulse voltammetry (DPV) is employed for alpha-fetoprotein (AFP) detection, demonstrating a linear relationship between AFP concentration and peak current in antibody-immobilization-based and label-free detection strategies [60].

In a second study from 2022, CV was used to investigate chemical reactions involving electron transfer during electrode surface modification with electrochemically exfoliated GQDs and CD44 antibodies. CV results show a gradual decrease in peak current with each modification step, confirming the effectiveness of antibody immobilization and biolayer formation on the electrode. Furthermore, EIS was employed to study changes in charge transfer resistance (RCT) in modified electrodes, demonstrating a decrease in RCT with GQD incorporation and a subsequent increase with CD44 antibody immobilization and bovine serum albumin (BSA) adsorption. Lastly, differential pulse voltammetry (DPV) was utilized for ultrasensitive detection of the CD44 antigen, showing a linear response over a wide concentration range and high sensitivity of the developed biosensor [78].

The same year, a mobile device integrated graphene oxide quantum-dot-based electrochemical biosensor was developed to detect miR-141 as a pancreatic cancer biomarker. This study employed a combination of CV, DPV, and EIS techniques. CV analysis revealed a concentration-dependent decrease in current values as miR-141 concentration increased, suggesting the binding of miR-141 to the sensor and hindrance of electron transfer at the surface. DPV measurements further validated the biosensor’s sensitivity, demonstrating a proportional drop in current values with increasing miR-141 concentration. Additionally, EIS analysis provided insights into changes in electron transfer resistance, confirming the effectiveness of the biosensor preparation and supporting the findings obtained from CV analysis [127].

In 2023, an ultrasensitive electrochemical biosensor was developed to simultaneously detect microRNA-21 and microRNA-155 based on the specific interaction of antimonide quantum dots with RNA. DPV experiments were conducted to scrutinize each step of the modified electrode, confirming the validity of electrochemically amplified signals and the microRNA complexes modified with single-walled carbon nanotubes (SWCNTs). DPV results revealed significantly higher oxidation peaks of NB (−6.4 V) and MB (−0.3 V) in the single-stranded RNA complex. Moreover, a notable reduction in the oxidation peaks of NB and MB was observed after the addition of complementary microRNAs, indicating facile desorption of the hybridization target from the antimonide quantum dot interface. Sequential characterization of the sensor assembly was performed via cyclic voltammetry (CV). Higher Fe(CN)_6_^3−/4^ oxidation peaks were observed on the SWCNT-modified electrode, followed by the self-assembly of the AMQDs/ssRNA reactive complex on the SWCNT/SPCE surface. Subsequently, the detection of microRNA-21 and microRNA-155 exhibited increased oxidation peaks of Fe(CN)_6_^3−/4^, signifying the successful orientation of the microRNAs [128].

Electrochemical DNA biosensors with a dual-signal amplification strategy were also designed for highly sensitive HPV 16 detection. The utilization of cyclic voltammetry enabled the characterization of each operational step, demonstrating improved electron transfer efficiency due to the self-assembled APTES film and a gradual decline in oxidation peaks owing to DNA hybridization and the addition of DNA probes [63]. These techniques enabled the validation of amplified signals, studying electrode modification, optimizing electron transfer, and assessing biosensor sensitivity. While not new, the strategic application of these techniques is crucial to achieving sensitive and specific detection of relevant biomolecules in medical and biotechnological applications. The characteristics of the studies that used DPV for electrochemical characterization are found in Table 5.

#### 4.2.4. Square Wave Voltammetry (SWV) 

Square wave voltammetry (SWV) is a highly versatile electrochemical technique that combines the best features of various pulse voltammetric methods. It integrates the background suppression and sensitivity of differential pulse voltammetry, the diagnostic value of normal pulse voltammetry, and the ability to interrogate electrochemical products akin to reverse pulse voltammetry directly. Typically performed at a stationary electrode, SWV employs a unique waveform involving measurement cycles without diffusion layer renewal between cycles. This technique provides detailed insights into the kinetics and thermodynamics of electrochemical reactions and is executed using computer-controlled potentiostatic systems for precise and efficient data analysis [121,123,129].

In 2018, SWV was performed within a potential range of −1.0 to −0.3 V at a scan rate of 0.1 V/s, while EIS was analyzed in a frequency range of 10^6^ to 0.1 Hz with an AC amplitude of 5 mV. Both techniques were used in a 0.1 M KCl solution containing Fe(CN)_6_^3−/4^. Finally, the electrochemical response was determined through differential pulse voltammetry (DPV) in 0.1 M PBS (pH 7.0) with a pulse amplitude and width of 50 mV and 0.05 s, respectively. These techniques enabled the sensitive detection of MCF-7 cells, demonstrating the efficacy of the proposed biosensor for diagnostic applications [80].

In 2021, a notable study introduced an electrochemical platform incorporating gold nanoparticles, GQDs, and graphene oxide films. CV responses of the biosensor were extensively assessed across various scan rates in a KCl solution containing Fe(CN)_6_^3−/4^, demonstrating diffusion-controlled electron-transfer processes.

Moreover, SWV elucidated distinct peaks corresponding to redox indicators, facilitating concurrent multiple microRNA detections. Under optimal assay conditions, SWV peak currents exhibited correlations with miRNA-21, miRNA-155, and miRNA-210 concentrations, enabling ultrasensitive detection with wide linear ranges and low detection limits (LODs). Additionally, EIS revealed enhancements in charge transfer resistance (Rct) upon modification of electrode surfaces, particularly with the incorporation of gold nanoparticles, graphene quantum dots, and graphene oxide, thereby improving electrochemical performance for microRNA detection [59]. Another notable study in 2021 presented an electrochemical biosensor utilizing antimonide for the ultrasensitive detection of microRNA-21. Characterization through CV and EIS elucidated the stepwise assembly process, showcasing the successful construction of the microRNA biosensor. Under optimal parameters, SWV unveiled distinct electrochemical signals attributed to the oxidation of Cd2+ on the biosensor surface, enabling the precise determination of microRNA-21 concentrations with exceptional sensitivity and low detection limits [65].

In 2023, the electrochemical immunosensor for ultra-low detection of human papillomavirus biomarkers for cervical cancer underwent CV analyses to assess its performance. The SWV method was employed to monitor the reaction between the immunosensors and different concentrations of the HPV-16 L1 antigen. The continuous suppression of redox peaks upon increasing antigen concentration indicated excellent complexation between the antibody and antigen. Despite the broader peak response of the onion-like carbon (OLC)-based immunosensor, the polyacrylonitrile (OLC-PAN)-based counterpart exhibited superior linearity, enabling its use for HPV-16 L1 antigen detection across various concentrations [83]. The characteristics of the studies that used SWV for electrochemical characterization are found in Table 6.

#### 4.2.5. Chronoamperometry (CA) 

CA is an electrochemical technique that records the electric current as a function of time during an electrochemical experiment. This method involves applying a constant potential to an electrode and measuring the current flowing in response to the electrochemical reaction at the electrode–solution interface. Chronoamperometry helps study the kinetics of electrochemical reactions, the formation of electrochemical products, and the dynamics of electrochemical interfaces. It allows for investigating how electric currents vary over time and provides valuable insights into the underlying electrochemical processes [121,123].

In 2019, various electrochemical techniques, including CV, DPV, EIS, and CA, were employed to validate the progression of electrode modification. The CV method recorded cyclic voltammograms at a scan rate of 100 mV/s in a 2 mM catechol solution to assess the redox reactions at each modification step.

Modifying GCE with Bio AuNP and Bio AuNP/CD exhibited increased oxidation peak currents, attributed to the unique properties of CDs enhancing surface-to-volume ratio and quantum size effects. DPV analyses validated the CV results, while EIS illustrated changes in electron transfer resistance after each modification step. CA facilitated the rapid and reproducible registration of signals post-addition of CD, Bio AuNP, aptamer sequence, antigen, probe DNA, and target oligonucleotide [130]. These findings underscore the utility of electrochemical methods in monitoring and optimizing the fabrication of biosensors, paving the way for enhanced analytical capabilities in biochemical sensing applications.

The biosensor demonstrated high selectivity, showing no significant decrease in peak current after hybridization with mismatch targets or non-complementary DNA sequences. It exhibited excellent reproducibility, with relative standard deviations (RSDs) of 0.014 and 0.012 for the DNA sensor and aptasensor. Negligible changes in peak current confirmed stability after one week of storage at 4 °C. The sensor also achieved low detection limits of 1.5 pM for DNA and 0.26 pg mL^−1^ for the aptamer sensor. Additionally, it maintained linear relationships between peak currents and the logarithm of target DNA and CEA antigen concentrations across a wide range [114].

#### 4.2.6. Linear Sweep Voltammetry (LSV) 

LSV is an electrochemical technique that understands a system’s behavior by sweeping the potential linearly with time while recording the resulting current. Compared to other methods, LSV provides a more efficient and comprehensive way to analyze electrochemical reactions, offering insights into the presence of different species and the kinetics of reactions. It is conducted with sweep rates ranging from 10 mV/s to 106 V/s and is commonly used to obtain current-potential curves, aiding in the qualitative and quantitative analysis of electrochemical processes [121,123]. 

In a 2021 study, various electrochemical techniques were employed to optimize the sensor’s performance. The CV technique was utilized for the electrodeposition of GQDs onto the precleaned GCE, with the cycle number optimized to achieve the best thickness of GQDs on the GCE surface. EIS was subsequently employed to confirm the CV results. Gold nanoparticles (AuNPs) were electrochemically synthesized using the CA technique after GQD deposition to enhance the conductivity of the modified electrode. The linear sweep voltammetry (LSV) reduction technique was used to optimize CHA voltammograms and determine the appropriate potential for AuNP deposition. Additionally, the optimum volume of biotinylated antibodies immobilized on the modified electrode was determined using LSV oxidation results. Finally, SWV was employed to generate a calibration curve correlating the electrical current with the number of immobilized cells on the modified electrode, showcasing the sensitivity and utility of the developed electrochemical cytosensor [74].

The biosensor demonstrates unique specificity for the biorecognition of KG1a cells, indicating high selectivity. It exhibits excellent reproducibility, with a relative standard deviation (RSD) of 1.5% for 50 repeated measurements at a concentration of 1 cell/mL, highlighting its remarkable stability. The sensitivity of the cytosensor is reflected in its LOD, determined to be 1 cell/mL. Furthermore, it shows good linearity with a linear dynamic range from 1 to 25 cells/mL [64].

It is evident in the articles that most of these studies employ two or more electrochemical techniques. Electrochemical impedance spectroscopy (EIS) and cyclic voltammetry (CV) are the most used techniques. Combining these methodologies has become a common practice due to their ability to comprehensively evaluate biosensors’ electrochemical and surface properties. This trend underscores the importance of supporting findings with complementary approaches, contributing to a more thorough and reliable characterization of biosensors for cancer detection.

### 4.3. Techniques for the Characterization of Optical Biosensors

The evolution in the characterization of optical biosensors from 2019 to 2023 highlights significant advances in biomedical detection, starting with photoluminescence spectroscopy to study the emission properties of quantum dots and progressing to sophisticated techniques such as confocal microscopy to visualize the interaction between complexes and tumor cells [52,131]. Innovations include the use of fluorescence emission spectra and Raman spectroscopy to analyze the interaction with microRNA and carbon nanostructures in cells, expanding to a combination of methods such as UV–VIS, FTIR, SEM, TEM, and CLSM to evaluate the optical, structural, and morphological properties of the biosensors [110,132,133,134]. This progress culminated in the implementation of CLSM in 2023 to detail cellular uptake and intracellular trafficking of nanoparticles, highlighting the importance of comprehensive characterization for developing and applying optical biosensors in medical diagnostics, particularly in cancer detection [135]. Table 7 summarizes the key characteristics and methodologies of studies utilizing optical characterization techniques, emphasizing the range and evolution of methods applied over this period.

## 5. Regulatory Challenges in the Clinical Translation of Carbon-Based Biosensors

The clinical translation of electrochemical and optical carbon dots (CDs) and glassy carbon biosensors represents a significant advancement in the diagnostic technology. These biosensors offer high sensitivity, specificity, and rapid detection capabilities, making them suitable for various applications in clinical diagnostics. However, their integration into existing diagnostic frameworks faces regulatory challenges and requires careful consideration of their compatibility with the current tools.

One of the primary concerns in the clinical deployment of biosensors is ensuring their biocompatibility and safety. Carbon-based nanomaterials, while offering high biocompatibility in in vitro models, must demonstrate nontoxic behavior in biological environments. This concern is underscored by the unique properties of carbon nanomaterials, which have been shown to exhibit varying degrees of toxicity depending on their structure, size, and surface modifications. For instance, studies have indicated that graphene oxide (GO) is less toxic than other carbon-based nanomaterials, highlighting the importance of material selection in biosensor applications [136,137].

Furthermore, the interaction between carbon nanomaterials and biological systems is complex and can lead to immunological responses. Research has shown that the physicochemical properties of these materials significantly influence their biological interactions, which can either promote biocompatibility or induce toxicity [138,139]. In addition to biocompatibility, the design of biosensors that incorporate carbon-based nanomaterials must consider their behavior in vivo. The systemic toxicity of these materials can vary widely, necessitating rigorous testing to ensure that they do not pose any risks when deployed in biological environments [139,140].

Manufacturing consistency and standardization presents additional regulatory hurdles. Variations in the synthesis and functionalization of carbon dots and glassy carbon can significantly affect biosensor performance. Standardizing the production processes to ensure reproducibility and uniformity across different batches is essential for regulatory approval [141]. The need for consistency extends to the characterization of nanoscale properties, which must adhere to well-defined regulatory specifications to ensure reliability in clinical applications [142].

Another critical requirement for regulatory compliance is clinical validation. Biosensors must undergo rigorous trials to demonstrate diagnostic accuracy, sensitivity, and specificity under diverse clinical conditions. For example, biosensors for cancer diagnostics must reliably detect biomarkers with minimal false positives or negatives across various patient populations [143]. This necessitates robust clinical evidence, which often demands time-consuming and resource-intensive studies.

Finally, regulatory pathways must account for the integration of these technologies into the existing diagnostic systems. Carbon-based biosensors must be compatible with current platforms and workflows to ensure seamless implementation in clinical settings. Additionally, biosensors that involve the collection and analysis of patient data must comply with strict data protection regulations, such as GDPR or HIPAA, addressing privacy and ethical concerns [144].

Addressing these regulatory challenges is paramount for unlocking the clinical potential of carbon-based biosensors. A coordinated effort among researchers, manufacturers, and regulatory bodies is needed to ensure that these innovative tools meet safety, efficacy, and ethical standards, thereby facilitating their adoption in clinical practice.

## 6. Summary and Conclusions

The review highlights recent advancements in electrochemical and optical biosensors utilizing CDs and GC for cancer detection. The studies demonstrate that these nanostructured materials have greatly improved biosensor performance in terms of sensitivity, selectivity, and stability. A comparison between optical and electrochemical approaches reveals their respective advantages and challenges. Optical biosensors are recognized for their high sensitivity, rapid response, and user-friendly nature, while electrochemical biosensors offer better selectivity, robustness, and the potential for miniaturization. The combination of the unique properties of CDs and GC has led to the development of hybrid biosensors with enhanced detection characteristics.

In summary, the literature indicates that CD- and GC-based biosensors hold promise for the early and accurate detection of various types of cancer. Compared to conventional methods, these devices significantly improve key parameters, such as sensitivity, selectivity, and stability. However, every approach is deemed superior, and the choice of biosensor type will depend on the application’s specific requirements. The review also underscores the significance of electrochemical techniques, such as electrochemical impedance spectroscopy (EIS), cyclic voltammetry (CV), and differential pulse voltammetry (DPV), in comprehensively characterizing biosensors. These techniques provide valuable information on electron transfer properties, electrode surface modification, and overall device performance, thereby contributing to developing and optimizing advanced biosensing platforms. Continued innovation in the synthesis and modification of CDs and GC, in addition to the development of hybrid strategies combining the strengths of optical and electrochemical approaches, is expected to drive further significant advances in the early detection and monitoring of cancer through highly efficient and reliable biosensors.

## 7. Future Perspectives

Future research in developing electrochemical and optical CDs and GC biosensors should center on leveraging these technologies for more precise and early cancer detection and potential therapeutic applications. Integrating these biosensors with novel biomarkers, particularly those based on single genes or proteins, can facilitate precise detection tailored to individual cancer types.

To propel this field forward, scientists should prioritize the development of multifunctional platforms capable of simultaneously measuring multiple outputs from a single biosensor. This approach enhances diagnostic accuracy and paves the way for personalized treatment strategies. Further research is essential to improve the stability and reduce the cost of nanostructure fabrication, particularly in functionalizing metal layers with 2D materials, antibodies, aptamers, and proteins. Overcoming these challenges will be pivotal for commercializing and widespread adoption of these biosensors in clinical settings.

Moreover, exploring the combination of optical and electrochemical detection mechanisms within a single device could yield hybrid biosensors with superior performance characteristics. Such innovations could revolutionize cancer diagnostics, enabling earlier detection and more effective monitoring of treatment responses. Future investigations should also focus on synthesizing and modifying CDs and GC, exploring their potential in detection, targeted drug delivery, and real-time monitoring of therapeutic outcomes. By addressing these areas, the scientific community can significantly contribute to developing next-generation biosensors that offer robust, cost-effective cancer detection and treatment solutions.

## Figures and Tables

**Figure 1 micromachines-16-00139-f001:**
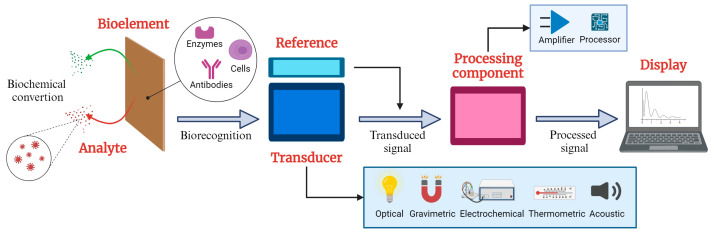
Schematic diagram of a typical biosensor (created with BioRender^®^ 2024).

**Figure 2 micromachines-16-00139-f002:**
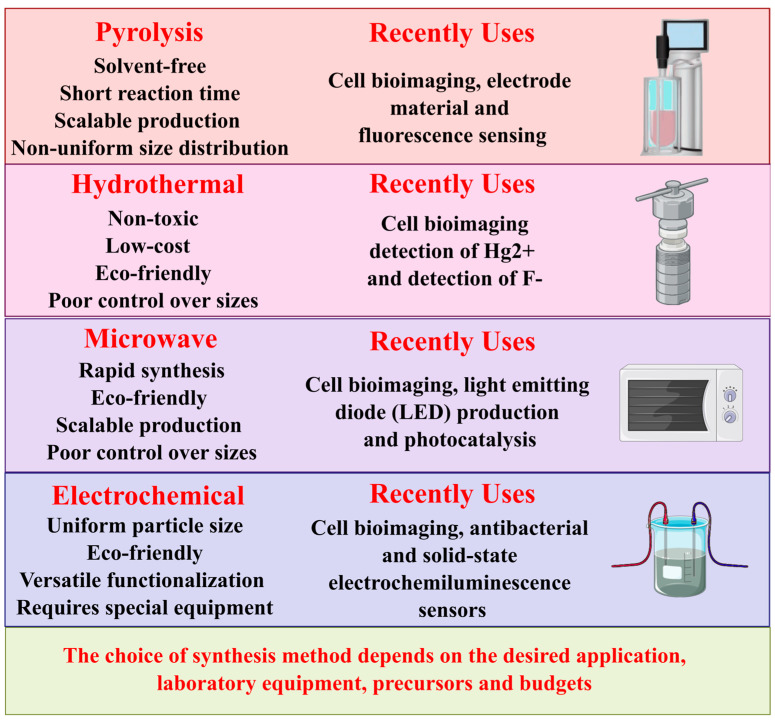
Synthesis techniques for carbon-based nanomaterials: recent applications and key features.

**Figure 3 micromachines-16-00139-f003:**
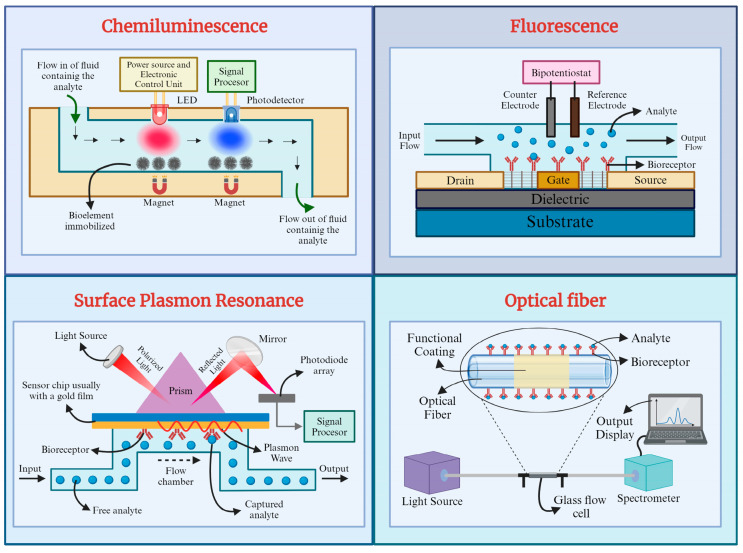
Schematic diagrams of chemiluminescence, fluorescence, surface plasmon resonance (SPR), and optical fiber biosensors (created with BioRender^®^ 2024).

**Table 2 micromachines-16-00139-t002:** Types of biosensors and skin cancer.

Biosensor Material	Biomarker	Type of Biosensor	Detection Mechanism	Detection Conditions	Year	Ref.
CDs and molecular-beacon-based	MicroRNA-21	Non-enzymatic	Fluorescence (FRET-based)	50 mM PBS buffer (pH 7.4); 20 uL CD-MB-BHQ1 conjugate (6 uM) mixed with microRNA-21 (up no 2 uM) in a total volume of 200 uL; 20 min incubation.	2019	[110]
AuPt nanoparticles, VG sheets, GCE	Alpha-fetoprotein (AFP)	Enzymatic	Electrochemical	MO/CNT-Au/Ab2 with surface of Ag-Ab1-AuPt-VG/GCE and incubated. Electrochemical measurements were then conducted using electrode in a PBS solution at different pH values.	2019	[60]
Bimetallic CoCu-ZIF nanosheets and MXene-derived carbon dots	B16-F10 cells	Non-enzymatic	Electrochemical	Normal mice cells (L929 cells) as interferents, as well as other cancer cells (MCF-7, 4T1, K7M2, CT26), cancer markers (PSA, EGFR, AFP, VEGF, Mb, Tn, IgG), and protein (10 pg·mL^−1^).	2021	[117]
GQDs modified with Au nanoparticles and CoP-BNF to modify the GCE	HER2 (human epidermal growth factor)	Non-enzymatic	Electrochemical	NPs/HB5, and lastly, GCP/SNGQDs@AuNPs/HB5/HER2. The XPS spectrum of modified GC plates was recorded from 0 to 1200 eV.	2022	[68]
AMNFs, CDs	microRNA-21	Non-enzymatic	Electrochemical	Optimized incubation time and ssRNA-Cd^2+^-CD concentrations for efficient detection, with 100 pM identified as the optimal concentration for targeting microRNA-21.	2023	[79]

**Table 3 micromachines-16-00139-t003:** Characteristics of studies employing EIS for electrochemical characterization.

Selectivity	Reproducibility	Stability	Sensitivity	Linearity	Year	Ref.
GCE||CNS–Au@S-GQDs/Ang-2 biosensor high selectivity for glioma cells, and minimal response to interfering cell MCF-7 and MCF-10.	Precision of 3.8% detecting glioma cells, measurements across 5 different electrodes.	GCE||CNS–Au@S-GQDs/Ang-2 intact over 4 weeks of storage at 4 °C, with recovery rates 84–94%.	The detection limit (LOD) of 40 cells mL^−1^.	Good linearity (*R*^2^ = 0.972) detecting glioma cells (100–50,000 cells mL^−1^).	2021	[77]
Strong selectivity for the target DNA sequence (T) and non-complementary sequences due to the covalent conjugation of DNA probes on CDs.	N/A	*	LOD was 2 aM.	Linear fluorescence intensity correlation with BRCA1 concentration range of 0.16 fM–6.8 fM.	2022	[47]
High selectivity for miRNA-27a-3p detection, unaffected by miRNA-205, miRNA-155, and miRNA-221 concentrations.	Good reproducibility in detecting gastric cancer, with higher miRNA-27a-3p expression levels, recoveries between 89.1% and 104.2%.	*	*	Intensity correlates positively with miRNA-27a-3p 1 fM to 10 nM, $R^{2}$ of 0.9919, Mo_2_TiC_2_ QDs and SnS_2_ nanosheets.	2023	[79]

* Mentioned but not quantified. *R*^2^ is the linear relationship between the sample and electrode.

**Table 4 micromachines-16-00139-t004:** Characteristics of studies employing EIS and CV for electrochemical characterization.

Selectivity	Reproducibility	Stability	Sensitivity	Linearity	Year	Ref.
Large difference in EIS between cancerous cells (HeLa and MCF-7) and normal cells (MCF-10 and bEnd.3).	N/A	N/A	LODs obtained for HeLa and MCF-7 were 246 and 367 cells mL^−1^, respectively.	Linear range of 5 × 10^2^–10^5^ cells mL^−1^.	2018	[125]
The electrochemical signals of HeLa and Hct116 cells were far lower than K562 tumor cells; high specificity.	Three different concentrations of K562 cells. RSD assays ranged from 5.26% to 7.22%.	N/A	LOD of 60 cells mL^−1^ (S/N = 3).	Linear correlation high *R*^2^ = 0.9986.	2019	[62]
Detected UBE2C in breast cancer cell MCF-7 extract, outperforming conventional ELISA.	Five different sensors with 0.05 mg mL^−1^ UBE2C an RSD value of 3.51%. And testing 0.05 mg mL^−1^ UBE2C 5 times, observed RSD of 3.11%.	Evaluated by storing PBS at 4 °C. After 4 weeks, retained 86% of its initial response 0.05 mg mL^−1^ UBE2C.	LOD and limit of quantification (LOQ) of 7.907 pg mL^−1^ and 26.356 pg mL^−1^.	Linear correlation *R*^2^ = 0.9914 in the range of 500 pg mL^−1^ to 5 mg mL^−1^.	2019	[69]
Minimal change in impedance exposed to potential interferences like AFP, Tau protein, Hb, L-Cys, and L-glu at 10 ng mL^−1^. A significant increase impedance when CEA.	Measured concentrations of CEA ranging from 10.4 to 11.5 ng mL^−1^ and AVG of 10.9 ng mL^−1^. Low RSD was 6.8%.	Recovery rate of 98 ± 3% after 10 days of storage. Even after 20 days of storage, retained 87 ± 4% of its original activity.	LOD of 0.01 ng mL^−1^.	Linear range 0.5–1000 (ng mL^−1^).	2019	[126]
The specific detection of PTK7 (0.1 pg·mL^−1^) and the ability to distinguish between different types of cells and protein markers indicated good selectivity.	Five measurements independent Apt/CoCu-ZIF@CDs/AEs toward B16-F10 cells with 3 concentrations of 5 × 10^2^, 1 × 10^3^, and 5 × 10^3^ cells mL^−1^.	Storing Apt/CoCu-ZIF@CDs/AE in a refrigerator (4 °C) for 15 days and continuously detecting B16-F10 cells daily by EIS.	LOD was deduced to be 33 cells∙mL^−1^.	The B16-F10 concentration range from 1 × 10^2^ cells∙mL^−1^ to 1 × 10^5^ cells∙mL^−1^.	2021	[117]
Detected HER2 in human serum samples, showcasing its selectivity amidst diverse serum components.	Through repeated measurements at various HER2 concentrations, RSD consistently below 8%.	After 4 days of storage, retained 90% of its performance ability.	LOD of GCE/PPy@SNGQDs/CoPc (6)/HB5 at 0.00141 ng/mL, and highest LOD for GCE/CoPc (2)/HB5, 0.647 ng/mL.	Linear range 1–10 ng/mL.	2021	[64]
Both the antibody (Trastuzumab) and aptamer (HB5) probes showed competitive performance in capturing HER2.	Excellent reproducibility of both sensors, with RSDs of less than 2% for all electrodes at a HER2 concentration of 5 ng/mL.	Over 3 days stored at 4 °C, the Rct values were comparable to those of the initial day; retention rates: immunosensor 97% and aptasensor 96%.	LOD range from 0.0112 ng/mL to 0.0489 ng/mL.	N/A	2022	[68]
GCE/AuNPs/CoTAPc (8)/HB5 and GCE/SNGQDs/CoTAPc(seq.) (6)/HB5 for HER2 detection in human serum.	Aptasensors: excellent reproducibility at various concentrations of HER2.	GCE/AuNPs (4)/CoTAPc (8)/HB5/HER2 probe showed the highest stability (0.56% RSD) over the 96 h at 4 °C.	LOD: achieved by the GCE/AuNPs (4)/HB5 probe (0.006 ng/mL), highest LOD with GCE/SNGQDs (2)/HB5 probe (0.29 ng/mL).	GCE/CeO2NPs (3)/HB5, GCE/SNGQDs(π) CoTAPc (5)/HB5, and GCE/AuNPs/CoTAPc (8)/HB5, *R*^2^ > 0.98.	2023	[70]

*R*^2^ is the linear relationship between the sample and electrode.

**Table 5 micromachines-16-00139-t005:** Characteristics of studies employing DPV for electrochemical characterization.

Selectivity	Reproducibility	Stability	Sensitivity	Linearity	Year	Ref.
High selectivity in both methods (less than 7% variation), in the presence of interfering substances.	Higher reproducibility in label-free method, RSD of 1.22% compared to the sandwich-type method with 2.07% RSD.	After 1 month of storage at 4 °C, the sandwich-type method showed a peak current decrease of 1.5%, while the label-free method decreased by 4.0%.	The sandwich-type strategy offered higher sensitivity with an LOD of 0.7 fg mL^−1^. Label-free method with a LODof 0.9 fg mL^−1^.	The sandwich-type had linear range from 1 fg mL^−1^ to 100 ng mL^−1^, and a higher correlation coefficient (*R*^2^ = 0.996) compared to the label-free strategy (*R*^2^ = 0.990).	2019	[60]
High selectivity for CD44 detection, various interfering analytes such as PSA, CEA, SCC-9 cells, IgG, MDA, HEK-293-T, and dopamine at 50.0 pg/mL.	Low RSD of 5.55% for 5 consecutive differential pulse voltammetry (DPV) scans.	N/A	LOD of 2.11 fg/mL in PBS. In spiked serum samples with a LOD of 2.71 fg/mL.	Range from 0.1 pg/mL to 100.0 ng/mL. It maintained a linear response from 1.0 pg/mL to 100.0 ng/mL.	2022	[78]
Higher response to miR-141 than to miR-21 at equivalent concentrations.	N/A	N/A	LOD of 0.091 pM and a LOQ of 0.27 pM for miR-141 detection.	Linear range spanning from 2.3 to 6.1 nM for miR-141 detection.	2022	[127]
Minimal interference from mismatched single-stranded RNAs; specificity for microRNA detection.	Low RSD of 3.6% across multiple measurements of microRNA-21 and microRNA-155.	The biosensor achieved high recovery rates (98.4% to 105%) and low RSDs (<3.1%).	Rapid detection times (80 min) with ultralow detection limits of 64 aM and 89 aM for microRNA-21 and microRNA-155.	Linear detection capabilities for microRNA-21 and microRNA-155 ranging from 0 to 1 pM.	2023	[128]
High specificity for target HPV 16 DNA; minimal interference from other DNA sequences.	N/A	A minimal decrease in current response 1.008% after 7 days and 2.420% after 14 days, its reliability over time.	LOD of 1.731 × 10^−16^ mol/L.	Linear response ranges from 1.0 × 10^−13^ mol/L to 1.0 × 10^−5^ mol/L, *R*^2^ = 0.99232.	2023	[63]

*R*^2^ is the linear relationship between the sample and electrode.

**Table 6 micromachines-16-00139-t006:** Characteristics of studies employing SWV for electrochemical characterization.

Selectivity	Reproducibility	Stability	Sensitivity	Linearity	Year	Ref.
High selectivity towards MCF-7 cells, current change compared to other cell types.	RSD of less than 4.6% across five electrodes.	Over 90.6% of the initial response remained constant after 14 days of storage at 4 °C.	LOD of 80 cells mL^−1^.	Detecting MCF-7 cells within the range of 0 to 1.0 × 10^6^ cells mL^−1^, *R*^2^ = 0.9868.	2018	[80]
Significant binding miRNA sequences compared to close sequences and non-complementary miRNAs.	RSDs range from 4.51% to 9.43% across 15 fabricated electrodes for each target miRNA.	After 3 weeks of storage at 4 °C, retained 84.3% to 89.5% of the initial response values.	LODs range from 0.04 fM to 0.33 fM.	Wide linear dynamic ranges from 0.001 to 1000 pM.	2021	[59]
Lower oxidation peak currents for complementary targets compared to mismatched and non-complementary targets.	RSD of 2.6% when detecting microRNA-21 at a concentration of 10 fM.	After 12 days, retained approximately 101.2% of the original signal at 100 fM.	LOD of 21 aM.	Linear relationship between peak current and microRNA, determination, *R*^2^ = 0.994).	2021	[65]
Specificity towards HPV-16 L1 antigen over native ovalbumin protein.	Detected the antigen and underwent a stripping process using glycine HCl solution (pH 2.8) for 5 min. It was then reused to detect the HPV-16 L1 antigen.	Repetitive regeneration detection steps, maintaining its functionality even after storage at 4 °C for 7 days.	Excellent sensitivity (>5.2 μA/log ([HPV-16 L1, fg/mL]), and low LOD of 1.83 fg/mL (32.7 aM) and 0.61 fg/mL (10.9 aM) for OLC-PAN and OLC-based immunosensors.	Two electrode platforms were used: OLC and OLC-PAN composites. Wide linear concentration range (1.95 fg/mL to 6.25 ng/mL).	2023	[83]

*R*^2^ is the linear relationship between the sample and electrode.

**Table 7 micromachines-16-00139-t007:** Characteristics of studies employing optical characterization.

Selectivity	Reproducibility	Stability	Sensitivity	Linearity	Year	Ref.
No significant fluorescence signals were detected; EpCAM with BSA and IgG at the same concentration.	Consistent results testing different cell lines, including Hep G2, A549, and HEK293 cells.	N/A	LOD was 1.19 nM.	The linear range was between 2 and 64 nM.	2019	[131]
High specificity microRNA-21, from single mismatch mutants and scrambled sequences.	N/A	Stability regarding its structural integrity, performance, and physical properties.	LOD wasf.3 nM microRNA-21.	Linearity: microRNA-21 (between 5 and 160 nM) with neat linearity, y = 1.1407x + 58.37.	2019	[110]
Detecting cell-selective therapeutic functionality towards HeLa cervical cancer.	Good reproducibility. No observable discrepancies over 2 months.	Stability in terms of material properties and Raman behavior; timeframe of 2 months.	Achieved a SERS enhancement factor of 10^7^, high compared to Raman enhancement factors.	*	2020	[133]
Recording the fluorescence, including target microRNA-155 and microRNA-21. Did not influence the detection of microRNA-21.	N/A	They exhibited good mechanical, physical, and fluorescence properties.	LOD was 0.03 fM.	Range of 0.1 to 125 fM for GA-CDs-CH and NB-CDs-CH hydrogels, and 0.1 to 26.3 fM for B-CDs-CH hydrogels.	2020	[132]
Strong ECL signals (HeLa and MCF-7) due to their high metabolism and abundant release of H_2_O_2_; only a weak signal was detected with normal cells.	N/A	Stored at 4 °C for further use after fabrication, indicating a standard practice to maintain the stability of biosensors.	Notable increase in the ECL corresponding to the increased concentration of cancer cells (HeLa and MCF-7).	*	2020	[52]
High specificity for the HE4 biomarker.	N/A	Stability in operation, time-saving characteristics; good robustness in the analysis samples.	LOD was low as 2.3 pM. Also achieved 196 cells mL^−1^ for ovarian cancer cells.	HE4-positive ovarian cancer cells, range of 1.02 × 10^4^ to 2.56 × 10^6^ cells mL^−1^.	2021	[134]
It selectively induced blue solid fluorescence in cancer cells but not in normal cells.	N/A	*	*	N/A	2023	[135]

* Mentioned but not quantified.

## Data Availability

Not applicable.

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
