# Peer review of "Electrochemical and Optical Carbon Dots and Glassy Carbon Biosensors: A Review on Their Development and Applications in Early Cancer Detection"

_micromachines, 2025, doi:10.3390/mi16020139_

Round 1

Reviewer 1 Report

Comments and Suggestions for Authors

The manuscript gives a comprehensive overview of the development and application of carbon dots (CDs) and glassy carbon (GC) biosensors for early cancer detection. The authors have clearly articulated the significance of biosensors in oncology, emphasizing their electrochemical and optical properties, synthesis methods, and applications. The integration of detailed descriptions of biomarkers, characterization techniques, and biosensor development ensures a well-rounded narrative. The manuscript can be recommended for publication following the minor revisions noted below.

  1. Authors should expand on the potential clinical translation of these technologies, including regulatory challenges and integration with existing diagnostic tools.
  2. Author should Ensure uniformity in terms like "CDs" and "carbon dots" throughout the manuscript.
  3. Please correct typographical errors, e.g., "pf 7.4" should likely be "pH 7.4."
  4. Authors listed many tables result; they should consider adding more figures to enhance visual appeal of the manuscript. For example, radar charts or bar graph to illustrate improvements in key parameters like sensitivity and selectivity.

Author Response

Thank you very much for taking the time to review this manuscript. Please find the detailed responses below and the corresponding corrections in track changes in the re-submitted files.

  1. Authors should expand on the potential clinical translation of these technologies, including regulatory challenges and integration with existing diagnostic tools.

R/ Yes, thank you very much for your comments; we included a new section, No. 5 Regulatory Challenges in the Clinical Translation of Carbon-Based Biosensors, to improve the manuscript. P29, L1046.

  1. Author should Ensure uniformity in terms like "CDs" and "carbon dots" throughout the manuscript.

R/ Yes, we agree; all manuscripts were changed.

  1. Please correct typographical errors, e.g., "pf 7.4" should likely be "pH 7.4."

R/ Yes, you are right. Thank you very much for your comments. We corrected in the table 2, P16 L660.

  1. Authors listed many tables result; they should consider adding more figures to enhance visual appeal of the manuscript. For example, radar charts or bar graph to illustrate improvements in key parameters like sensitivity and selectivity.

R/ Yes, thank you very much for your comments; several corrections will be made. Figure 1 shows a Graphical Abstract with the chemical structures of CDs and GCs to improve the understanding of the review's objective. Figure 3 shows the synthesis techniques for carbon-based nanomaterials.

Reviewer 2 Report

Comments and Suggestions for Authors

The review summarized Electrochemical and Optical Carbon Quantum Dots and Glassy Carbon Biosensors in early cancer detection. The review can be considered for publication in "Micromachines" after revising the following questions. The comments are below.

1) In the introduction, the author should explain the progress of biosensors in recent years and cite more relevant articles.

2) In the introduction part, objective and extent of the review is not specified. As it described other biosensors thoroughly, it is confusing to get the purpose of the paper without mentioning.

3) In introduction, author only describe hybridization of Carbon dots (CDs) and glassy carbon (GC), draw possible chemical structures and include as a figure in manuscript.

4) In the introduction, the author should explain the progress of biosensors in recent years and cite more relevant articles.

ACS sensors, DOI: https://doi.org/10.1021/acssensors.4c02972

Biosensors and Bioelectronics, DOI: https://doi.org/10.1016/j.bios.2025.117134

Biosensors and Bioelectronics, DOI: https://doi.org/10.1016/j.bios.2024.116338

5) In section 2.0 “Development of Biosensors”, two techniques are explained with pros and cons, however which techniques is optimum and recently uses is need to be added.

6) In section 2.1, write values of temperature, pressure and voltage for hydrothermal synthesis, Microwave-assisted synthesis and electrochemical method of CDs.

7) In 2.2 Synthesis of Glassy Carbon, first write how many methods are available for synthesis and then continue with Despite the continued dominance of pyrolyzing phenol/formaldehyde (PFA) resins as the preferred method for…”

8) 3.1 Enzymatic and Non-Enzymatic Biosensors, if possible, add Limit of Detection (LOD) along with sensitivity of these sensor, to compare the results of these two sensors.

9) For a review, the author should consider using a diagram to summarize the content at the beginning or end to help readers understand the scope of the article.

10) The composition of the article needs to be adjusted. First the line number should be added. Second, there is excess blank at the beginning of each page. In addition, headings and content on different pages should be avoided.

11) The author should enrich the expression of the article and add more pictures in the article to improve the display quality of the article. 

Author Response

Thank you very much for taking the time to review this manuscript. Please find the detailed responses below and the corresponding corrections in track changes in the re-submitted files.

  1. In the introduction, the author should explain the progress of biosensors in recent years and cite more relevant articles.

R/ Thank you very much for your comments. The introduction was updated with references from the last three years, from refs six to ten, P2 Lines 61, 65, and 70.

  1. In the introduction part, objective and extent of the review is not specified. As it described other biosensors thoroughly, it is confusing to get the paper's purpose without mentioning.

R/ Yes, thank you for the corrections; we have included a paragraph in the abstract and a Graphical Abstract to improve understanding of the review's objective.

  1. In introduction, author only describe hybridization of Carbon dots (CDs) and glassy carbon (GC), draw possible chemical structures and include as a figure in manuscript.

R/ Thank you very much for your comments. We have included a Graphical Abstract with the chemical structures of CDs and GCs to improve the understanding of the review's objective.

  1. In the introduction, the author should explain the progress of biosensors in recent years and cite more relevant articles.

R/ Yes, you are right—the same answer as in #1; we included the suggestion references in the introduction, P2 L51 ref. 3; P3 L61 ref. 6; P3 L61 ref. 7.

  1. In section 2.0 “Development of Biosensors”, two techniques are explained with pros and cons, however which techniques is optimum and recently uses is need to be added.

R/ Thank you very much for your comments. In our revised manuscript, we have included the most used fabrication methods, emphasizing that the selection of a specific process is highly dependent on the physicochemical and purity characteristics required for the biosensor and the intended applications. These requirements ultimately determine which synthesis method offers the greatest advantages for a given scenario.

  1. In section 2.1, write values of temperature, pressure and voltage for hydrothermal synthesis, Microwave-assisted synthesis and electrochemical method of CDs.

R/ Thank you very much for your comments. We have included in section 2.1 the most commonly used temperature, power and pressure ranges for hydrothermal and microwave-assisted synthesis. P 5; L167 and L170.

  1. In 2.2 “Synthesis of Glassy Carbon”, first write how many methods are available for synthesis and then continue with “Despite the continued dominance of pyrolyzing phenol/formaldehyde (PFA) resins as the preferred method for…”

R/ Thank you very much for your observations and recommendations. We have corrected the text in the Synthesis of Glassy Carbon section.

  1. 3.1 Enzymatic and Non-Enzymatic Biosensors, if possible, add Limit of Detection (LOD) along with sensitivity of these sensor to compare the results of these two sensors.

R/ Thank you very much for your comments. We have included in section 3.1 a paragraph to compare the detection limit (LOD) of the biosensors studied to improve the understanding of the text and the tables. P16, L598.

  1. For a review, the author should consider using a diagram to summarize the content at the beginning or end to help readers understand the scope of the article.

R/ Thank you very much for your observations and recommendations. We have included a Graphical Abstract with the chemical structures of CDs and GCs to improve the understanding of the review's objective.

     10. The composition of the article needs to be adjusted. First the line number should be added. Second, there is excess blank at the beginning of each page. In addition, headings and content on different pages should be avoided.

R/ Thank you very much for your observations and recommendations. Line numbering was included in the manuscript; spaces in the header and extra lines were removed; the text was adjusted so that the heading and descriptions were not separated on different pages.

      11. The author should enrich the expression of the article and add more pictures in the article to improve the display quality of the article.

R/ Yes, thank you very much; several corrections will be made. Figure 1 shows a Graphical Abstract with the chemical structures of CDs and GCs to improve the understanding of the review's objective. Figure 3 shows the synthesis techniques for carbon-based nanomaterials.

Reviewer 3 Report

Comments and Suggestions for Authors

The authors offer a detailed review on carbon quantum dots and glassy carbon biosensors of cancer: they focus on biosensor production, types of sensors (enzymatic-nonenzymatic, optical, electrochemical, etc.), sensor characterization (selectivity, sensitivity, stability, linearity and so on). However, it would be very good point if the authors also write about disadvantages of each type of cancer biosensor: for example, low reproducibility, cross-reactions, time consumability, etc. For this reason, the manuscript needs major revision: this is a good, thorough piece of work, but there are some minor points that need to be addressed:

1)      P. 1: “a sensitive bioelement that recognizes the target analyte (microorganisms, cells,antibodies, or enzymes” – I guess small compounds like dopamine or glucose can also be an analyte;

2)      P. 3 “[20], [21], [22]” – Citations should be merged here and elsewhere in the text;

3)      P. 3: “Gold nanoparticles, quantum dots, carbon nanotubes, graphene, and other nanostructured material” - The authors totally ignore to mention metal nanocluster-based biosensors, which should be mentioned at least in the Introduction: for example, see detection of cancer related proteins (https://www.sciencedirect.com/science/article/abs/pii/S138614252300481X, https://pubs.rsc.org/en/content/articlelanding/2020/an/d0an01538e) and low molecular weight compounds like DOPA and dopamine as analytes (https://www.sciencedirect.com/science/article/abs/pii/S138614252300495X);

4)      P. 8: “In a 2021 study titled …” – There is no need to add the full titles of cited papers: simple citation is sufficient;

5)      P. 8: All abbreviations used in Table 1 should be deciphered somewhere in the main text;

6)      P. 11: “Fluorescence-Based Optical Biosensors” – the amount of cited literature in this section is absolutely insufficient;

7)      P. 15 - The authors do not mention pterins as low molecular weight cancer biomarkers, which is a vast field of studies: for example, see https://www.sciencedirect.com/science/article/abs/pii/S0731708516301005,

 https://www.mdpi.com/2075-4418/10/9/612. Pterins can be mentioned as biomarkers of bladder cancer;

8)      The glutathione as a cancer biomarker is totally ignored also: please, see https://www.scirp.org/journal/paperinformation?paperid=114810;

9)      The amount of text in Tables 2-6 should be reduced to make the tables more concise.

Author Response

Thank you very much for taking the time to review this manuscript. Please find the detailed responses below and the corresponding corrections in track changes in the re-submitted files.

  1. “a sensitive bioelement that recognizes the target analyte (microorganisms, cells, antibodies, or enzymes” – I guess small compounds like dopamine or glucose can also be an analyte;

R/ Thank you very much for your observations and recommendations. We have made some significant changes. We have corrected the introduction to improve understanding of the paragraph, "The analyte refers to the analyzed substance, ranging from small molecules, such as glucose and dopamine [2], to macromolecules, including proteins, nucleic acids, and polysaccharides [3]." 

  1. “[20], [21], [22]” – Citations should be merged here and elsewhere in the text;

R/ Thank you very much for your observations and recommendations. We have made the suggested changes to the citation format throughout the text, e.g., P2, L70. 

  1. “Gold nanoparticles, quantum dots, carbon nanotubes, graphene, and other nanostructured material” - The authors totally ignore to mention metal nanocluster-based biosensors, which should be mentioned at least in the Introduction: for example, see detection of cancer related proteins (https://www.sciencedirect.com/science/article/abs/pii/S138614252300481X, https://pubs.rsc.org/en/content/articlelanding/2020/an/d0an01538e) and low molecular weight compounds like DOPA and dopamine as analytes (https://www.sciencedirect.com/science/article/abs/pii/S138614252300495X);

R/ Thank you very much for your comments. The introduction and development of biosensors sections were updated with references from the last five years, references two P2 Line 50, references twelve P3 Line 70, and references thirty-eight P4 Line 128. 

  1. “In a 2021 study titled …” – There is no need to add the full titles of cited papers: simple citation is sufficient;

R/ Thank you very much for your observations and recommendations. We have removed the titles of the articles in which they are referenced throughout the text and improved the manuscript description. 

  1. All abbreviations used in Table 1 should be deciphered somewhere in the main text;

R/ Thank you very much for your observations and recommendations. We have included the description of the abbreviations used in the table taken from the bibliographical references to give you a better understanding of the content of the table. P10, Table 1. 

  1. “Fluorescence-Based Optical Biosensors” – the amount of cited literature in this section is absolutely insufficient;

R/ Thank you very much for your observations and recommendations. We have included in section 2.4.2.1. more updated references to improve the understanding of the text; the references were P12, L474 ref. 41, P12, L472 refs. 91 - 93. 

  1. The authors do not mention pterins as low molecular weight cancer biomarkers, which is a vast field of studies: for example, see https://www.sciencedirect.com/science/article/abs/pii/S0731708516301005,

 https://www.mdpi.com/2075-4418/10/9/612. Pterins can be mentioned as biomarkers of bladder cancer; 

R/ Thank you very much for your comments. In the definition of bioanalyte, we include any molecule intended for analysis, which also encompasses those with small molecular sizes. Regarding the case of biosensors or analytes related to cancer biosensing, the focus may often shift toward macromolecules with higher molecular weights. We have ensured to include references that also address the biosensing of small molecules. 

  1. The glutathione as a cancer biomarker is totally ignored also: please, see https://www.scirp.org/journal/paperinformation?paperid=114810;

R/ Thank you very much for your comments. We have included references related to the biomarker glutathione in the revises manuscript. 

  1. The amount of text in Tables 2-6 should be reduced to make the tables more concise.

A/ Yes, thank you very much; several corrections will be made. The text within the tables was reduced from 2 to 7, to make the information more concise. 

Round 2

Reviewer 3 Report

Comments and Suggestions for Authors

The authors performed extensive manuscript editing in short-terms. The comments to the reviewers are point-by-point and sufficient. Now the manuscript is thoroughly and well-organized. The reference list is fully equipped. I recommend the manuscript for publication in Micromachines.